# DOI: OFFLINE DIVERSITY MAXIMIZATION UNDER IMITATION CONSTRAINTS

## ABSTRACT

There has been significant recent progress in the area of unsupervised skill discovery, utilizing various information-theoretic objectives as measures of diversity. Despite these advances, challenges remain: current methods require significant online interaction, fail to leverage vast amounts of available task-agnostic data and typically lack a quantitative measure of skill utility. We address these challenges by proposing a principled offline algorithm for unsupervised skill discovery that, in addition to maximizing diversity, ensures that each learned skill imitates state-only expert demonstrations to a certain degree. Our main analytical contribution is to connect Fenchel duality, reinforcement learning, and unsupervised skill discovery to maximize a mutual information objective subject to KL-divergence state occupancy constraints. Furthermore, we demonstrate the effectiveness of our method on the standard offline benchmark D4RL and on a custom offline dataset collected from a 12-DoF quadruped robot for which the policies trained in simulation transfer well to the real robotic system.[1]

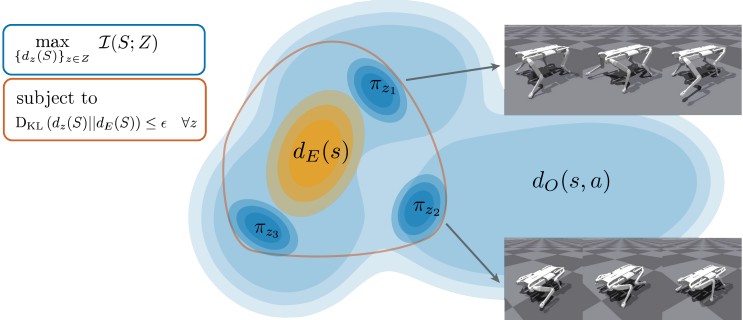

Figure 1: Diverse Offline Imitation (DOI) maximizes a variational lower bound on the mutual information between latent skills $z$ and states $s$ visited by associated skill-conditioned policies $\pi_z$, subject to a KL-divergence constraint to limit the deviation of the state occupancy $d_z(s)$ of each latent skill $z$ from that of an expert $d_E(s)$.

## 1 INTRODUCTION

Recent advancements in reinforcement learning (RL) have included substantial progress in unsupervised skill discovery, aiming to empower autonomous agents with the capability to acquire a diverse set of skills directly from their environment, without relying on predefined human-engineered rewards or demonstrations. These methods have the potential to revolutionize the way RL agents learn to solve complex tasks. The growing interest in unsupervised skill discovery has led to various approaches, typically rooted in information-theoretic concepts, including empowerment (Klyubin et al., 2005; Mohamed and Jimenez Rezende, 2015; Eysenbach et al., 2019), information bottleneck (Tishby et al., 1999; Goyal et al., 2019; Kim et al., 2021a) and information gain (Houthooft

---

[1]Project website with videos: https://tinyurl.com/diversity-via-duality

et al., 2016; Strouse et al., 2022; Park and Levine, 2023). Despite these advancements, there remains a significant challenge. Current methods demand substantial online interaction with the environment, making exploration in high-dimensional state-action spaces inefficient. Although Zahavy et al. (2022) introduced constraints to enhance skill performance and narrow the exploration space by incentivizing diverse skills to meet a certain utility measure, their approach does not eliminate the need for considerable online interaction with the environment. Meanwhile, there have been significant recent advances in large-scale data collection (Rob, 2020; Walke et al., 2023; Brohan et al., 2023) and in the development of scalable and sample-efficient offline RL algorithms that leverage diverse behaviors of pre-collected experience. However, these approaches struggle with well-known challenges, including off-policy evaluation and the out-of-distribution problem, which have been studied extensively in previous work (Levine et al., 2020; Prudencio et al., 2022).

In this work, we address the aforementioned challenges by introducing a novel problem formulation and complementing it with the first principled "offline" RL algorithm for unsupervised skill discovery that, in addition to maximizing diversity, ensures that each learned skill imitates state-only expert demonstrations to a certain degree. More specifically, we consider a problem formulation with two datasets: a large one with diverse state-action demonstrations and another much smaller one with state-only expert demonstrations. This setting is particularly valuable in robotics scenarios where expert demonstrations are limited and the domain of the expert may be different from that of the agent, such as in human demonstrations. Another potential application is to enhance the realism of computer games by creating an immersive experience of interacting with non-player characters, each behaving in a slightly different style, while all partially imitating the behavior of a human expert.

We formulate the problem as a Constrained Markov Decision Process (CMDP) (Altman, 1999; Szepesvári, 2020) that seeks to maximize diversity through a mutual information objective, subject to Kullback-Leibler (KL) divergence state occupancy constraints ensuring that each skill imitates state expert demonstrations to a certain degree. The resulting CMDP has convex objective and constraints, making the optimization problem intractable. We adopt a tractable relaxation approach consisting of an alternating scheme that maximizes a variational lower bound on mutual information, and to handle the constraints it applies Lagrange relaxation. Our method, Diverse Offline Imitation (DOI), overcomes the off-policy evaluation by leveraging the Fenchel-Rockafellar duality in RL (Nachum and Dai, 2020; Kim et al., 2022; Ma et al., 2022) to connect a dual optimal value solution (computed using offline samples) with primal optimal state-action occupancy ratios. These ratios serve as importance weights for offline training of a skill-conditioned policy, skill-discriminator, KL-divergence estimators, and Lagrange multipliers. We demonstrate the effectiveness of our method on the standard offline benchmark D4RL (Fu et al., 2020) and on a custom offline dataset collected from a 12-DoF quadruped robot Solo12 (Léziart et al., 2021). In addition, we show that DOI on simulation data transfers well to a real robot system.

## 2 RELATED WORK

In the context of skill discovery Achiam et al. (2018) and Campos et al. (2020) showed that methods like DIAYN (Eysenbach et al., 2019) can struggle to learn large numbers of skills and have a poor coverage of the state space. Strouse et al. (2022) observed that when a novel state is visited, the discriminator lacks sufficient training data to accurately classify skills, which results in a low intrinsic reward for exploration. They address this by introducing an information gain objective (involving an ensemble of discriminators) as a bonus term. Kim et al. (2021b) gave a skill discovery approach based on an information bottleneck that leads to disentangled and interpretable skill representations. Park et al. (2022; 2023) proposed a Lipschitz-constrained skill discovery method based on a distance-maximizing and controllability-aware distance function to overcome the bias toward static skills and to allow the agent to learn complex and far-reaching behaviors. Sharma et al. (2020) developed a method that simultaneously discovers predictable skills and learns their dynamics. In a follow-up work, Park and Levine (2023) addresses the problem of errors in predictive models by learning a transformed MDP, whose action space contains only easy to model and predictable actions. These works provide RL algorithms for unsupervised skill discovery that require *online* interaction with the environment and do not impose utility measures on the learned skills. In contrast, DOI gives a principled *offline* algorithm for maximizing diversity under imitation constraints.

A large body of research has focused on successor features (Dayan, 1993; Barreto et al., 2016), a powerful technique in RL for transfer of knowledge across tasks by capturing environmental dynamics, particularly promising for skill discovery when coupled with variational intrinsic motivation (Gregor et al., 2017; Barreto et al., 2018; Hansen et al., 2020) to enhance feature controllability, generalization, and task inference. In contrast to our work, these approaches do not impose performance constraints on the learned skills. Zahavy et al. (2022) cast the task of learning diverse skills, each achieving a near-optimal performance with respect to a given reward, into a constrained MDP setting with a physics-inspired diversity objective based on a minimum $\ell_2$ distance between the successor features of different skills. However, this approach requires significant *online* interaction with the environment to learn the skills.

Numerous practical algorithms for offline RL have been proposed (Levine et al., 2020; Prudencio et al., 2022), including methods based on advantage-weighted behavioral cloning (Nair et al., 2020; Wang et al., 2020), conservative strategies to stay close to the original data distribution (Kumar et al., 2020; Cheng et al., 2022) and using only on-data samples (Kostrikov et al., 2022; Xu et al., 2023). While these methods excel at learning a policy that maximizes a fixed reward, they are not directly applicable in our setting, which has a non-stationary reward that depends on: i) the log-likelihood of a skill discriminator, and ii) Lagrange multipliers. In addition, these techniques cannot be used to i) train a skill discriminator and ii) estimate a KL divergence offline.

Naive importance sampling approaches for off-policy estimation are known to suffer from unbounded variance in the infinite horizon setting, a problem known in the literature as "the curse of horizon". Liu et al. (2018); Mousavi et al. (2020) addressed this challenge by providing theoretical foundations and a principled off-policy algorithm, using a backward Bellman operator, that avoids exploding variance by applying importance sampling to state-visitation distributions, and by providing practical solutions in Reproducing Kernel Hilbert Spaces. An alternative research direction in off-policy estimation, referred to as "Distribution Correction Estimation (DICE)", has introduced innovative techniques, with Nachum et al. (2019a) mitigating variance with importance sampling, Nachum et al. (2019b) enabling policy gradient from off-policy data without importance weighting, Kim et al. (2022) stabilizing offline imitation learning with imperfect demonstrations, Zhang et al. (2020) improving density ratio estimation, Dai et al. (2020) providing high-confidence off-policy evaluation. Subsequently, Xu et al. (2021) applied this approach to offline RL and demonstrated its effectiveness in continuous control tasks. Our work uses a DICE-based off-policy approach similar to OptiDICE (Lee et al., 2021; 2022) for estimating importance ratios, while considering a constrained formulation with a mutual information objective and KL-divergence imitation constraints.

## 3 PRELIMINARIES

We utilize the framework of Markov decision processes (MDPs) (Puterman, 2014), where an MDP is defined by the tuple $(\mathcal{S}, \mathcal{A}, \mathcal{R}, \mathcal{P}, \rho_0, \gamma)$ denoting the state space, action space, reward mapping $\mathcal{R} : \mathcal{S} \times \mathcal{A} \mapsto \mathbb{R}$, stochastic transition kernel $\mathcal{P}(s'|s, a)$, initial state distribution $\rho_0(s)$ and discount factor $\gamma$. A policy $\pi : \mathcal{S} \mapsto \Delta(\mathcal{A})$ defines a probability distribution over the action space $\mathcal{A}$ conditioned on the state, where $\Delta(\cdot)$ stands for the probability simplex.

Given a policy $\pi$, the corresponding state-action occupancy measure $d^\pi(s, a)$ is defined by $(1 - \gamma)\sum_{t=0}^{\infty} \gamma^t \Pr[s_t = s, a_t = a \,|\, s_0 \sim \rho_0, a_t \sim \pi(\cdot|s_t), s_{t+1} \sim \mathcal{P}(\cdot|s_t, a_t)]$ and its associated state occupancy $d^\pi(s)$ is given by marginalizing over the action space $\sum_{a \in \mathcal{A}} d^\pi(s, a)$.

In the skill discovery setting, $z \sim p(Z)$ denotes a fixed latent skill on which we condition a policy $\pi_z : S \times Z \mapsto \Delta(\mathcal{A})$. We will treat $p(Z)$ as a categorical distribution over a discrete set $Z$ of $|Z|$ many distinct indicator vectors in $\mathbb{R}^{|Z|}$. The skill-conditioned policy $\pi_z$ induces a state occupancy denoted by $d_z(s) := d^{\pi_z}(s)$, and when it is clear from the context we will refer to $d_z(s)$ as a "skill".

We consider an offline setting with access to the following datasets: i) $\mathcal{D}_E$ sampled from an expert state occupancy $d_E(S)$; and ii) $\mathcal{D}_O$ sampled from a state-action occupancy $d_O(S, A)$ generated by a mixture of behaviors. Our analysis makes use of the following coverage assumption on state occupancies.

**Assumption 3.1** (Expert coverage). *We assume that $d_E(s) > 0$ implies $d_O(s) > 0$.*

## 4 METHOD

Given an expert and a coverage dataset as above, we aim to solve *offline* the constrained optimization problem

$$\max_{\{d_z(S)\}_{z \in Z}} \mathcal{I}(S; Z) \tag{1}$$

$$\text{subject to} \quad \mathrm{D_{KL}}\left(d_z(S)||d_E(S)\right) \le \epsilon \quad \forall z, \tag{2}$$

where $\mathcal{I}(S; Z)$ denotes the mutual information between states and skills. Henceforth, we shall make use of color coding to highlight the diversity signal in blue and the imitation signal in orange. The preceding problem formulation and our algorithmic framework can be easily extended to capture: i) objectives in (1) that combine conditional mutual information (c.f. DADS in (Sharma et al., 2020)) and information gain (c.f. DISDAIN in (Strouse et al., 2022)); and ii) general $f$-divergence constraints in (2), see Nachum and Dai (2020); Ma et al. (2022). We leave the study of these variants for future work.

Since maximizing the mutual information is generally intractable, in line with previous work (Eysenbach et al., 2019) we assume that the latent skills are sampled uniformly at random, i.e., $p(z) = \frac{1}{|Z|}$, and as a trackable surrogate we consider instead the following variational lower bound

$$\mathcal{I}\left(S; Z\right) \ge \mathbb{E}_{p(z), d_z(s)}\left[\log q(z|s)\right] + \mathcal{H}\left(p(z)\right) = \sum_z \mathbb{E}_{d_z(s)}\left[\frac{\log\left(|Z|q(z|s)\right)}{|Z|}\right]. \tag{3}$$

Here with $q(z|s)$ we denote a skill-discriminator tasked with distinguishing between latent skills.

Ma et al. (2022) proposed an offline algorithm (SMODICE) that on input an expert dataset $\mathcal{D}_E \sim d_E(S)$ and a coverage dataset $\mathcal{D}_O \sim d_O(S, A)$ such that $\mathcal{D}_E \subset \text{States}[\mathcal{D}_O]$, trains a policy $\pi_{\widetilde{E}}$ which optimizes the problem

$$\min_\pi \mathrm{D_{KL}}\left(d^\pi(S)||d_E(S)\right), \tag{4}$$

and also outputs ratios $\eta_{\widetilde{E}}(s, a) = d_{\pi_{\widetilde{E}}}(s, a)/d_O(s, a)$ for every state-action pair $(s, a) \in \mathcal{D}_O$.

An important observation is that the state constraints (2) can be reduced to state-action constraints, by training an expert policy $\pi_{\widetilde{E}}$, which optimizes eq. (4). More specifically, for each latent skill $z$ we replace the state constraint (2) with the following state-action constraint

$$\mathrm{D_{KL}}\left(d_z(S, A)||d_{\widetilde{E}}(S, A)\right) \le \epsilon, \tag{5}$$

where $d_{\widetilde{E}}(s, a)$ denotes the state-action occupancy $d_{\pi_{\widetilde{E}}}(s, a)$ induced by the expert policy $\pi_{\widetilde{E}}$.

We focus on a reduction of CMDPs to MDPs using gradient-based techniques, known as Lagrangian methods (Borkar, 2005; Bhatnagar and Lakshmanan, 2012; Tessler et al., 2019). In contrast to prior work on CMDP, which has focused primarily on linear objectives and constraints, we consider the nonlinear setting with convex objectives and constraints. More specifically, we seek to maximize the right-hand side of eq. (3) subject to eq. (5). Solving this problem is equivalent to

$$\max_{\substack{d_z(s,a) \\ q(z|s)}} \min_{\lambda \ge 0} \sum_z \mathbb{E}_{d_z(s)}\left[\frac{\log\left(|Z|q(z|s)\right)}{|Z|}\right] + \sum_z \lambda_z\left[\epsilon - \mathrm{D_{KL}}\left(d_z(S, A)||d_{\widetilde{E}}(S, A)\right)\right], \tag{6}$$

where with $\lambda_z$ we denote the Lagrange multiplier corresponding to latent skill $z$.

### 4.1 APPROXIMATION SCHEME

We use a popular heuristic, known in the literature as *alternating optimization*, to approximately compute a local optimum of Problem (6). More precisely, the method alternates between optimizing each model while holding all others fixed, and iteratively refines the solution until convergence is reached or a stopping criterion is met. Furthermore, as we can guarantee in practice that the Lagrange multipliers $\lambda$ are always positive, we consider Problem (6) with $\lambda > 0$, that is

$$\max_{\substack{d_z(s,a) \\ q(z|s)}} \min_{\lambda > 0} \sum_z \lambda_z\left\{\epsilon + \mathbb{E}_{d_z(s,a)}\left[R_z^\lambda(s, a)\right] - \mathrm{D_{KL}}\left(d_z(S, A)||d_O(S, A)\right)\right\}, \tag{7}$$

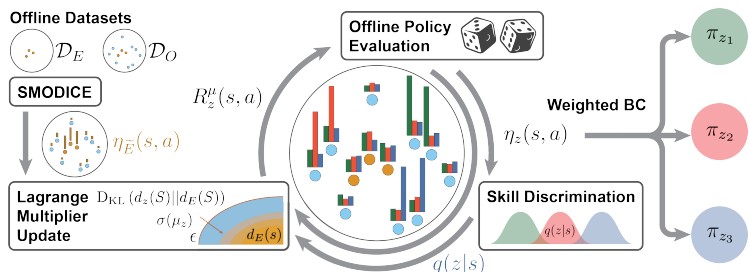

Figure 2: Illustration of Algorithm 1. We compute expert importance ratios $\eta_{\widetilde{E}}(s, a)$ by running SMODICE on the offline datasets $\mathcal{D}_E$ and $\mathcal{D}_O$. These expert ratios are then used in the alternating scheme described in Subsec. 4.1 to obtain the importance ratios $\eta_z(s, a)$ (with support in $\mathcal{D}_O$) for each skill $z$. Specifically, the skill-ratios $\eta_z(s, a)$ are computed by a DICE-like offline policy evaluation algorithm on input a reward $R_z^\mu(s, a)$ that balances skill diversity (skill-discriminator $q(z|s)$) and expert imitation (importance ratios $\eta_{\widetilde{E}}(s, a)$).

where

$$R_z^\lambda(s, a) := \underbrace{\frac{1}{\lambda_z}}_{\text{Constraint Violation}} \underbrace{\frac{\log\left(q(z|s)|Z|\right)}{|Z|}}_{\text{Skill Diversity}} + \underbrace{\log \eta_{\widetilde{E}}(s, a)}_{\text{Expert Imitation}}. \quad (8)$$

The reward in (8) is derived in Supp. B and relies on the following equality (see Supp. C.3) $\mathrm{D}_{\mathrm{KL}}(d_z(S, A)||d_{\widetilde{E}}(S, A)) = \mathrm{D}_{\mathrm{KL}}(d_z(S, A)||d_O(S, A)) - \mathbb{E}_{d_z(s,a)}[\log \frac{d_{\widetilde{E}}(s,a)}{d_O(s,a)}]$ and the definition of $\eta_{\widetilde{E}}(s, a) = d_{\widetilde{E}}(s, a)/d_O(s, a)$.

Intuitively, the reward $R_z^\lambda(s, a)$ balances between diversity and KL-closeness to the expert state-action occupancy. The Lagrange multiplier $\lambda_z$ scales down the log-likelihood of the skill-discriminator $q(z|s)$, effectively reducing the diversity signal, when the state-action occupancy $d_z(S, A)$ violates the KL-divergence constraint (5), and vice versa. Each term in the reward (8) involves a separate optimization procedure, which will be described in the next section.

## 4.2 APPROXIMATION PHASES

Using the alternating optimization scheme, Algorithm 1 decomposes into the following three optimization phases. In PHASE 1, we train a value function $V_z^\star$, ratios $\eta_z(s, a)$ and a skill-conditioned policy $\pi_z$. In PHASE 2, we train a skill-discriminator $q(z|s)$. Then in PHASE 3, we compute a KL constraint estimator $\phi_z$ and update accordingly the Lagrange multipliers $\lambda_z$. In addition, we perform a preprocessing phase to compute the expert ratios $\eta_{\widetilde{E}}(s, a)$ by invoking the SMODICE algorithm.

### 4.2.1 PHASE 1

With fixed skill-discriminator $q(z|s)$ and Lagrange multipliers $\lambda > 0$, Problem (7) becomes

$$\max_{\{d_z(s,a)\}_{z \in Z}} \sum_z \lambda_z \left\{ \mathbb{E}_{d_z(s,a)} \left[ R_z^\lambda(s, a) \right] - \mathrm{D}_{\mathrm{KL}} \left( d_z(S, A)||d_O(S, A) \right) \right\}, \quad (9)$$

or equivalently for every skill $z$:

$$\max_{d_z(s,a) \geq 0} \quad \mathbb{E}_{d_z(s,a)} \left[ R_z^\lambda(s, a) \right] - \mathrm{D}_{\mathrm{KL}} \left( d_z(S, A)||d_O(S, A) \right)$$

$$\text{subject to} \quad \sum_a d_z(s, a) = (1 - \gamma)\rho_0(s) + \gamma \mathcal{T}d(s) \quad \forall s, \quad (10)$$

where we denote with $\mathcal{T}$ the transition operator: $\mathcal{T}d(s') = \sum_{s,a} \mathcal{P}(s'|s, a)d(s, a)$.

**Assumption 4.1** (Strict Feasibility). *We assume there exists a solution such that the constraints (10) are satisfied and $d(s, a) > 0$ for all states-action pairs $(s, a) \in \mathcal{S} \times \mathcal{A}$.*

Using Lagrange duality, Assum. 4.1 (which implies strong duality) and the Fenchel conjugate (see Supp. A), Nachum and Dai (2020, Sec. 6) and Ma et al. (2022, Theorem 2) showed that Problem 10 shares the same optimal value as the following optimization problem

$$V^\star = \arg\min_{V(s)} (1 - \gamma)\mathbb{E}_{s \sim \rho_0} \left[ V(s) \right] + \log \mathbb{E}_{d_O(s,a)} \exp \left\{ R_z^\lambda(s, a) + \gamma \mathcal{T}V(s, a) - V(s) \right\}, \quad (11)$$

where $\mathcal{T}V(s,a) := \mathbb{E}_{\mathcal{P}(s'|s,a)}V(s')$. Moreover, the primal optimal solution is given by

$$\eta_z(s,a) := \frac{d_z^\star(s,a)}{d_O(s,a)} = \text{softmax}\left(R_z^\lambda(s,a) + \gamma\mathcal{T}V_z^\star(s,a) - V_z^\star(s)\right). \tag{12}$$

These ratios $\eta_z(s,a)$ are then used to design an offline importance-weighted sampling procedure that, for an arbitrary function $f$, satisfies

$$\mathbb{E}_{p(z)}\mathbb{E}_{d_z^\star(s,a)}[f(s,a,z)] = \mathbb{E}_{p(z)}\mathbb{E}_{d_O(s,a)}[\eta_z(s,a)f(s,a,z)]. \tag{13}$$

Afterwards, the optimal skill-conditioned policy $\pi_z^\star$ is trained offline using a weighted behavioral cloning, which is obtained by setting $f(s,a,z) = \log(\pi_z(a|s))$ and maximizing the RHS of eq. (13) over all skill-conditioned policies $\pi_z$. In practice, gradient descent is used for optimization.

### 4.2.2 PHASE 2

We now give an offline procedure for training a skill-discriminator $q(z|s)$, which takes as input ratios $\eta_z(s,a)$ of a skill-conditioned policy $\pi_z^\star$. The proof is presented in Supp. C.2.

**Lemma 4.2.** *Given ratios $\eta_z(s,a)$, using eq. (13) applied with $f(s,a,z) = \log(q(z|s))$, we can compute offline an optimal skill-discriminator $q^\star(z|s)$. In particular, we optimize by gradient descent the following optimization problem $\max_{q(z|s)} \mathbb{E}_{p(z)}\mathbb{E}_{d_O(s,a)}[\eta_z(s,a)\log(q(z|s))]$.*

The skill-conditioned policy $\pi_z^\star$ (PHASE 1) and the skill-discriminator $q^\star$ (PHASE 2), allow us to maximize *offline* the variational lower bound in eq. (3) and thus skill diversity. It remains to estimate possible constraint violations in eq. (5) and to update the Lagrange multipliers accordingly.

### 4.2.3 PHASE 3

With fixed skill-discriminator $q^\star(z|s)$ and skill-conditioned policy $\pi_z^\star(s)$, Problem (7) reduces to $\min_{\lambda>0}\sum_z \lambda_z\left[\epsilon - D_{\text{KL}}\left(d_z^\star(S,A)||d_{\widetilde{E}}(S,A)\right)\right]$. We will optimize the Lagrange multipliers by gradient descent. To this end, we now give an offline estimator of the KL-divergence term. The proof is presented in Supp. C.3.

**Lemma 4.3.** *Given skill-conditioned policy ratios $\eta_z(s,a)$ and expert ratios $\eta_{\widetilde{E}}(s,a)$, using eq. (13) applied with $f(s,a,z) = \log(\eta_z(s,a)/\eta_{\widetilde{E}}(s,a))$, we can compute offline an estimator of $D_{\text{KL}}\left(d_z^\star(S,A)||d_{\widetilde{E}}(S,A)\right)$ which is given by $\phi_z := \mathbb{E}_{d_O(s,a)}[\eta_z(s,a)\log(\eta_z(s,a)/\eta_{\widetilde{E}}(s,a))]$.*

We note that the ratios $\eta_z(s,a)$ and $\eta_{\widetilde{E}}(s,a)$ are computed only on state-action pairs within the offline dataset $\mathcal{D}_O$. Furthermore, in practice, we ensure that these ratios are strictly positive, so that the KL estimator $\phi_z$ is well defined and bounded.

## 5 ALGORITHM

Our optimization method consists of three phases, each of which optimizes a specific model and fixes the remaining ones. It is important to emphasize that in contrast to prior work, our problem formulation considers an optimization problem with constraints. Furthermore, the reward function in eq. (8) is non-stationary, since it depends on the bounded Lagrange multipliers that balance diversity ($\log q(z|s)$) and expert imitation ($\log \eta_{\widetilde{E}}(s,a)$). This has significant algorithmic implications, as it requires solving a sequence of standard RL problems, each of which admits offline policy evaluation.

To smooth the transition of the reward signal between successive iterations, we enforce a slow change of the Lagrange multipliers. More specifically, we use the technique of bounded Lagrange multipliers (Stooke et al., 2020; Zahavy et al., 2022), which applies a Sigmoid transformation $\lambda = \sigma(\mu)$ component-wise to unbounded variables $\mu \in \mathbb{R}^{|Z|}$, so that the effective reward is a convex combination of a diversity term and an expert imitation term. In practice, this transformation ensures that $\lambda > 0$. Hence, the reward for each latent skill $z$ becomes

$$R_z^\mu(s,a) := (1 - \sigma(\mu_z))\frac{\log\left(q^\star(z|s)|Z|\right)}{|Z|} + \sigma(\mu_z)\log\eta_{\widetilde{E}}(s,a). \tag{14}$$

We now present the resulting multi-phase optimization procedure in Algorithm 1. For a practical implementation, we leverage the power of neural networks and deep learning techniques for accurate

function approximation. More specifically, we train an expert policy $\pi_{\widetilde{E}}$, a skill-conditioned policy $\{\pi_z\}_{z \in Z}$ and a value function $\{V_z\}_{z \in Z}$. While practically convenient, this means that each phase of Algorithm 1 is only approximately solved. In practice, we do not solve the optimization problem to optimality in each phase, but rather perform a few gradient descent steps.

---

**Algorithm 1** Diverse Offline Imitation (DOI)

---

**Input:** a state-only expert dataset $\mathcal{D}_E \sim d_E(S)$ and a state-action offline dataset $\mathcal{D}_O \sim d_O(S, A)$
**Pre-compute** a state-discriminator $c^\star : \mathcal{S} \to (0, 1)$ via optimizing the following objective with the gradient penalty in (Gulrajani et al., 2017) $\min_c \mathbb{E}_{d_E(s)}[\log c(s)] + \mathbb{E}_{d_O(s)}[\log(1 - c(s))]$
Apply **Phase 1** with reward $R(s, a) = \log \frac{c^\star(s)}{1 - c^\star(s)}$ to compute ratios $\eta_{\widetilde{E}}(s, a) := \frac{d_{\widetilde{E}}(s,a)}{d_O(s,a)}$ for all $s, a \in \mathcal{D}_O$

**Repeat until convergence:**
    **Phase 1.** (Fixed Lagrange multipliers $\sigma(\mu)$ and skill-discriminator values $q^\star(z|s)$)
    **For** each latent skill $z$:
        compute a value function $V_z^\star$ optimizing eq. (11) with reward $R_z^\mu(s, a)$ in eq. (14)
        compute ratios $\eta_z(s, a) := \frac{d_z^\star(s,a)}{d_O(s,a)} = \text{softmax}\left(R_z^\mu(s, a) + \gamma \mathcal{T} V_z^\star(s, a) - V_z^\star(s)\right)$ for all $s, a \in \mathcal{D}$
        train a skill-conditioned policy $\pi_z^\star = \arg\max_{\pi_z} \mathbb{E}_{d_O(s,a)}[\eta_z(s, a) \log \pi_z(a|s)]$

    **Phase 2.** (Fixed ratios $\eta_z(s, a)$ and bounded Lagrange multipliers $\sigma(\mu)$)
    Train a skill-discriminator $q^\star = \arg\max_{q(\cdot|s)} \mathbb{E}_{p(z)} \mathbb{E}_{d_O(s,a)}[\eta_z(s, a) \log q(z|s)]$

    **Phase 3.** (Fixed ratios $\eta_{\widetilde{E}}(s, a)$ and $\eta_z(s, a)$)
    Compute for each latent skill $z$ an estimator $\phi_z := \mathbb{E}_{d_O(s,a)}[\eta_z(s, a) \log(\eta_z(s, a)/\eta_{\widetilde{E}}(s, a))]$
    Optimize the loss $\min_\mu \sum_z \sigma(\mu_z)(\epsilon - \phi_z)$

---

## 6 EXPERIMENTS

For evaluation of our method we consider 12 degree-of-freedom quadruped robot, SOLO12 (Grimminger et al., 2020), on a simple locomotion task in both *simulation* and the *real* system. We provide further evaluation on the ANT, WALKER2D, HALFCHEETAH and HOPPER environments from the D4RL benchmark (Fu et al., 2020).

For the SOLO12 evaluation we collected domain-randomized offline and expert data from simulation in the Isaac Gym (Makoviychuk et al., 2021) using saved checkpoints obtained by training the robot to track a certain velocity of the base with a version of DOMiNO (Zahavy et al., 2022). We defer the training procedure of the policies used for data collection to the Supp. E. The *expert dataset* was collected by using the best deterministic skill-conditioned policy from the last checkpoint of the training procedure, which was trained to track forward velocity only. In contrast, the *offline dataset* was acquired by employing stochastic policies gathered from various checkpoints throughout the training of the expert, featuring multiple latent skills. More than half of the *offline dataset* was collected by a random Gaussian policy. In line with previous approaches by Kim et al. (2022) and Ma et al. (2022), our practical implementation aims to fulfill the expert coverage Assum. 3.1. To achieve this, we create the coverage dataset $\mathcal{D}_O$ by adding a small number of expert trajectories to the offline dataset, resulting in an (unlabeled) expert fraction of 1/160 in $\mathcal{D}_O$. To ensure that our algorithm does not have access to labeled expert actions, we discard them from the expert dataset. The resulting *expert dataset* $\mathcal{D}_E$ is used to learn a state classifier, in order to compute the ratios $\eta_{\widetilde{E}}(s, a)$. We trained the policy for 350 steps, where each step involves the stages described in Sec. 5. In each stage, we execute 200 epochs of batched training over the data. For the computation of the skill-ratios $\eta_z(s, a)$, we choose a projection $\Pi$ of the expert state (see Supp. I) that yields 3-dimensional planar and angular velocities of the robot's base in the base frame.

We have found that fitting the skill-discriminator $q(z|s)$ is prone to collapse to the uniform distribution. To alleviate this issue, in addition to the variational lower bound objective (3), we add the DISDAIN information gain term, proposed in (Strouse et al., 2022). This bonus term is an entropy-based disagreement penalty that estimates the epistemic uncertainty of the skill-discriminator, and is implemented in practice by an ensemble of randomly initialized skill-discriminators. Due to the high initial disagreement on unvisited states, this intrinsic reward provides a strong exploration signal and leads to the discovery of more diverse behaviors. Intuitively, for states with small epistemic uncertainty, the skill-discriminator (averaged over the ensemble members) should reliably discrimi-

nate between latent skills, thus making the intrinsic reward of the skill-discriminator's log-likelihood more accurate. In all figures, we denote with $\text{DOI}^\epsilon$ the different constraint levels. We defer further experiment details to Supp. K.

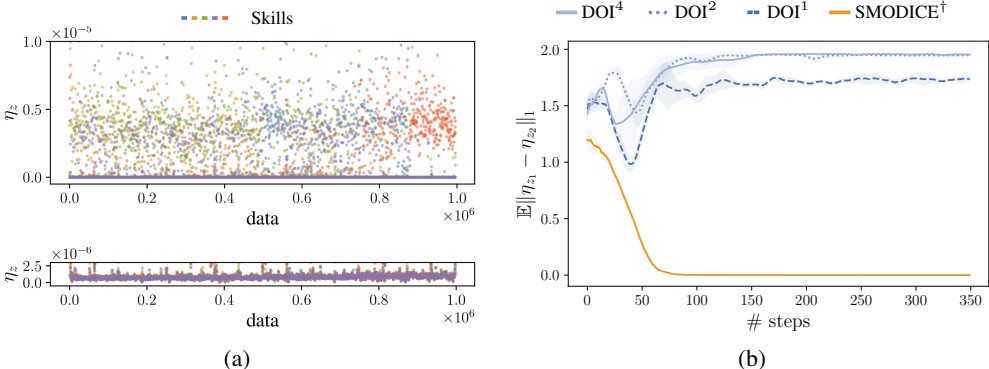

(a)

(b)

Figure 3: Data points separation by importance ratios $\eta_z(s, a)$, given different levels of $\epsilon$ in SOLO12. (a) Distribution of importance ratios $\eta_z(s, a)$ over the offline dataset $\mathcal{D}_O$ for different skills with $\text{DOI}^4$ ($\epsilon = 4$) (upper) and a skill-conditioned variant of SMODICE (lower). (b) Average $\ell_1$ distance of ratios $\eta_z$ belonging to different skills, depending on $\epsilon$. The higher the value of $\epsilon$, the greater the $\ell_1$ distance.

As a baseline, we consider a skill-conditioned variant of (Ma et al., 2022), denoted SMODICE[†], which does not have access to the skill-discriminator $q(z|s)$. This is equivalent to DOI with fixed $\sigma(\mu_z) = 1$ in the reward eq. (14). In Figure 3, we measure the state-action occupancy $d_z(s, a)$ for each latent skill $z$ through the proxy of importance ratios $\eta_z(s, a)$, for different values of $\epsilon$. As expected, a higher value of $\epsilon$ increases diversity, resulting in different importance ratios per skill for individual data points. We aggregate this difference by computing an average across different skills $\ell_1$ norm of the importance ratios $\mathbb{E}\|\eta_{z_i} - \eta_{z_j}\|_1$ and report it in Figure 3. We note that the looser the constraint (lighter color), the easier it is to "diversify" in the sense of $\eta_z$. In Figure 3a, we observe diversification across the dataset assignment to skills when using DOI, whereas training an ensemble of skills with only expert imitation reward (i.e., $\sigma(\mu_z) = 1$) collapses to nearly the same importance per skill per data point. Figure 3b shows the average $\ell_1$ distance between skill importance vectors $\eta_z$ over the dataset for $\epsilon \in \{0.0, 1.0, 2.0, 4.0\}$ (lighter color indicates higher $\epsilon$). Moreover, the tighter the constraint (smaller $\epsilon$), the smaller the difference between the different skill importance ratios.

We have further evaluated diversity on the Monte Carlo estimates of the expected successor feature of the initial state, based on 30 policy rollouts per skill. The $\gamma$-discounted successor features (SFs) for state $s$ are defined as $\psi_z(s) = \mathbb{E}_{d_z(s)}[\phi(s)]$, where $d_z(s)$ is the $\gamma$-discounted state occupancy for a skill policy $\pi_z$. With slight abuse of notation, we define $\psi_z = \mathbb{E}_{\rho_0(s)}[\psi_z(s)]$, the expected SFs over the initial state distribution. As a diversity metric, we take the average over different skills $\ell_2$ norm between SFs, i.e., $\mathbb{E}\|\psi_{z_1} - \psi_{z_2}\|_2$. The results are presented in Figure 4 and show an alignment with the proxy diversity metric, i.e. the separation of the data indicated by the importance ratios $\eta_z$ shows a higher distance between the expected SFs $\psi_z$. In terms of performance, DOI is able to achieve a forward velocity comparable to the expert (see Figure 4a) while diversifying the behavior in terms of base height $h$ (Figure 4b). We also observed that the multipliers $\sigma(\mu_z)$ are non-zero for all skills, indicating that the constraint is active. In addition, they stabilize at reasonable levels as training progresses, which we show in Supp. G for both the SOLO12 and ANT.

For D4RL environments, we consider the case where we have offline data generated from a random policy mixed with a small amount of expert trajectories.[2] Figure 5 shows the results for both the expected average SFs distance (Figure 5a) and the average importance ratio $\eta_z$ distance across skills (Figure 5b). We normalize the state feature $\phi(s)$ when comparing $\psi_z$ across environments in Figure 5a. As expected, there is a trade-off between the average skill return and the respective diversity metric across skills in most cases. Furthermore, the diversity distance that is more controllable by $\epsilon$ corresponds to the importance ratios $\eta_z$. This observation is in line with expectations, since $\eta_z$ is part of the constraint. Nonetheless, in Figure 5a we show that $\epsilon$ retains some controllability over

---

[2]The same setting was considered by Ma et al. (2022).

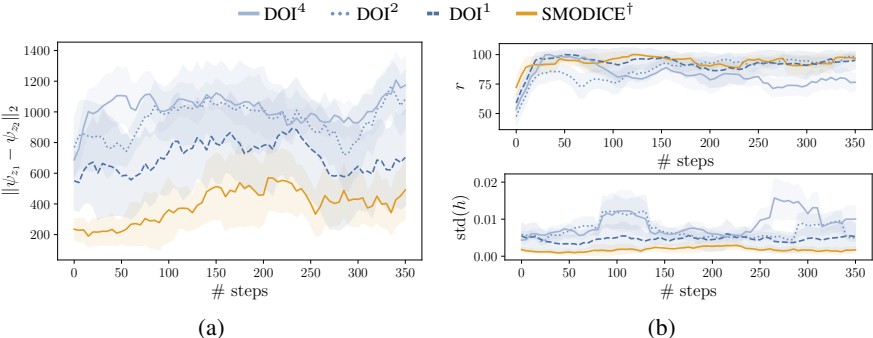

(a)

(b)

Figure 4: Average $\ell_2$ distance between MC estimated successor features $\psi_z$ of different skills (a), return $r$ as % of expert return and standard deviation of base height $\mathrm{std}_z(h)$ (b), depending on $\epsilon$ for the SOLO12.

diversity. The WALKER2D is particularly sensitive to relaxation of the occupancy constraint with respect to performance. We hypothesize that this is due to the fact that the space of policies that achieve a stable gait is very restrictive, resulting in a significant loss of task return for even slight skill diversification. In contrast, the ANT exhibits high stability, with multiple clusters achieving close to expert performance in terms of $r$. These results are also consistent with SMODICE expert policies used for computing $\eta_{\widetilde{E}}$ (see Supp. F).

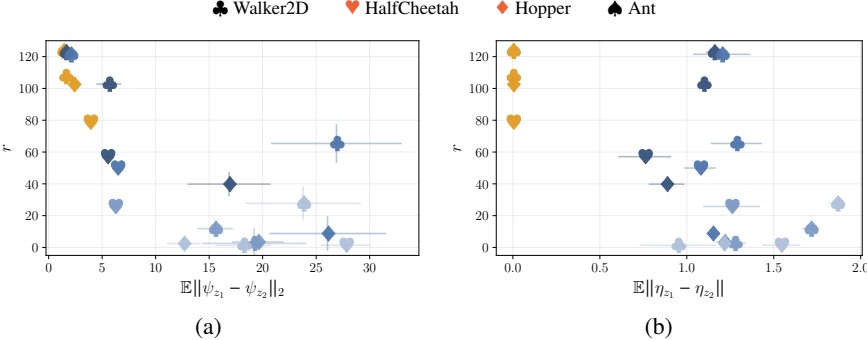

(a)

(b)

Figure 5: Results on D4RL environments with offline data collected from a random policy for $\epsilon = 0.0$, $0.5$, $1.0$, $2.0$, $4.0$. In figure (a) we observe the tradeoff between average skill return and average successor features distance over skills. In figure (b), we report the tradeoff w.r.t. average $\ell_1$ distance of importance ratios $\eta_z$.

## 7 CONCLUSION

We proposed DOI, a principled offline RL algorithm for unsupervised skill discovery that, in addition to maximizing diversity, ensures that each learned skill imitates state-only expert demonstrations to a certain degree. Our main analytical contribution is to connect Fenchel duality, reinforcement learning, and unsupervised skill discovery to maximize a mutual information objective subject to KL-divergence state occupancy constraints. We have shown that DOI can diversify offline policies for a 12-DoF quadruped robot (in simulation and in reality) and for several environments from the standard D4RL benchmark in terms of both $\ell_2$ distance of expected successor features and $\ell_1$ distance of importance ratios, which is visible from the data separation induced by $\eta_z(s, a)$ amongst skills. The importance ratio distance, computed offline, is a robust indicator of diversity, which aligns with the online Monte Carlo diversity metric of expected successor features. The resulting skill diversity naturally entails a trade-off in task performance. We can control the amount of diversity via a KL constraint level $\epsilon$, which ensures that different skills remain close to the expert in terms of state-action occupancy, which also indirectly controls task performance loss. A promising direction for future research is to impose constraints on the value function of each skill to ensure near-optimal task performance.

## 8 REPRODUCIBILITY

For implementation of DOI we have used the PyTorch autograd framework. For the SOLO12 training we made use of Isaac Gym for data collection and evaluation of the learned skill policies. For the D4RL experiments we evaluated the policies using the Mujoco v2.1 rigid body simulator. The training of the skill policies with evaluation and pre-training of the SMODICE expert ratios takes about 4 hours on an NVIDIA GeForce RTX 4080 graphics card with a batch size of 512. We plan on opensourcing the code and the SOLO12 data post conference acceptance. The SOLO12 robot has been developed as part of the Open Dynamic Robot Initiative (Grimminger et al., 2020), and a full assembly kit is available at a cheap price in order to reproduce the real system experiments from Supp. H.

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

# Supplementary for Diverse Offline Imitation Learning

## A    FENCHEL CONJUGATE

The Fenchel conjugate $f_\star$ of a function $f : \Omega \to \mathbb{R}$ is given by

$$f_\star(y) = \sup_{x \in \Omega} \langle x, y \rangle - f(x), \tag{S1}$$

where $\langle \cdot, \cdot \rangle$ denotes the inner product defined on a space $\Omega$. For any proper, convex and lower semi-continuous function $f$ the following duality statement holds $f_{\star\star} = f$, that is

$$f(x) = \sup_{y \in \Omega_\star} \langle x, y \rangle - f_\star(y), \tag{S2}$$

where $\Omega_\star$ denotes the domain of $f_\star$. For any probability distributions $p, q \in \Delta(S)$ with $p(s) > 0$ implying $q(s) > 0$, we define for convex continuous functions $f$ the family of $f$-divergences

$$\mathrm{D}_f(p||q) = \mathbb{E}_q \left[ f \left( \frac{p(x)}{q(x)} \right) \right]. \tag{S3}$$

The Fenchel conjugate of an $f$ divergence $\mathrm{D}_f(p||q)$ at a function $y(s) = p(s)/q(s)$ is, under certain conditions[3], given by

$$\mathrm{D}_{\star,f}(y) = \mathbb{E}_{q(s)} \left[ f_\star(y(s)) \right]. \tag{S4}$$

Furthermore, its maximizer satisfies

$$p^\star(s) = q(s) f'_\star(y(s)). \tag{S5}$$

In the important special case where $f(x) = x \log(x)$, we obtain the well-known Kullback-Leibler (KL) divergence

$$\mathrm{D}_{\mathrm{KL}}(p||q) = \sum_s p(s) \log \frac{p(s)}{q(s)}. \tag{S6}$$

The Fenchel conjugate $\mathrm{D}_{\star,\mathrm{KL}}$ of the KL-divergence at a function $y(s) = p(s)/q(s)$ has a closed-form (Boyd and Vandenberghe, 2004, Example 3.25)

$$\mathrm{D}_{\star,\mathrm{KL}}(y) = \log \mathbb{E}_{q(s)}[\exp y(s)], \tag{S7}$$

and its maximizer $p^\star$ satisfies

$$p^\star(s) = q(s)\mathrm{softmax}(y(s)). \tag{S8}$$

## B    LAGRANGE RELAXATION

The Lagrange relaxation is given by

$$\max_{d_z(s,a),q(z|s)} \min_{\lambda > 0} \sum_z \mathbb{E}_{d_z(s)} \left[ \frac{\log\left(|Z|q(z|s)\right)}{|Z|} \right] + \sum_z \lambda_z \left[ \epsilon - \mathrm{D}_{\mathrm{KL}}\left( d_z(S,A)||d_{\widetilde{E}}(S,A)\right) \right].$$

By combining Lem. C.5 and the definition of $\eta_{\widetilde{E}}(s,a) = \frac{d_{\widetilde{E}}(s,a)}{d_O(s,a)}$, we have

$$\mathrm{D}_{\mathrm{KL}}\left( d_z(S,A)||d_{\widetilde{E}}(S,A)\right) = \mathrm{D}_{\mathrm{KL}}\left( d_z(S,A)||d_O(S,A)\right) - \mathbb{E}_{d_z(s,a)} \left[ \log \eta_{\widetilde{E}}(s,a) \right]$$

and thus

$$\max_{d_z(s,a),q(z|s)} \min_{\lambda > 0} \sum_z \lambda_z \left[ \epsilon + \mathbb{E}_{d_z(s,a)} \left[ R_z^\lambda(s,a) \right] - \mathrm{D}_{\mathrm{KL}}\left( d_z(S,A)||d_O(S,A)\right) \right], \tag{S9}$$

where the reward is given by

$$R_z^\lambda(s,a) := \frac{\log\left(|Z|q(z|s)\right)}{\lambda_z|Z|} + \log \eta_{\widetilde{E}}(s,a).$$

---

[3] $f$ needs to satisfy certain regularity conditions (Dai et al., 2017)

## C  ALGORITHMIC PHASES

### C.1  VALUE FUNCTION TRAINING

With fixed skill-discriminator $q(z|s)$ and Lagrange multipliers $\lambda > 0$, the Problem S9 becomes:

$$\max_{\{d_z(s,a)\}_{z \in Z}} \sum_z \lambda_z \left\{ \mathbb{E}_{d_z(s,a)} \left[ R_z^\lambda(s,a) \right] - \mathrm{D}_{\mathrm{KL}} \left( d_z(s,a) || d_O(s,a) \right) \right\}$$

or equivalently for every skill $z$:

$$\begin{aligned} \max_{d_z(s,a) \geq 0} \quad & \mathbb{E}_{d_z(s,a)} \left[ R_z^\lambda(s,a) \right] - \mathrm{D}_{\mathrm{KL}} \left( d_z(S,A) || d_O(S,A) \right) \\ \text{s.t.} \quad & \sum_a d_z(s,a) = (1-\gamma)\rho_0(s) + \gamma \mathcal{T} d(s) \quad \forall s. \end{aligned} \tag{S10}$$

We note that the preceding problem formulation involves state-action occupancy.

The strict feasibility in Assumption 4.1 implies strong duality, and thus Problem (S10) shares the same optimal value as the following dual minimization problem (for details see (Nachum and Dai, 2020, Section 6) and (Ma et al., 2022, Theorem 2)):

$$\begin{aligned} V^\star = \quad & \arg\min_{V(s)} (1-\gamma)\mathbb{E}_{s \sim \rho_0} \left[ V(s) \right] \\ & + \log \mathbb{E}_{d^{\pi_O}(s,a)} \exp \left\{ R_z^\lambda(s,a) + \gamma \mathcal{T} V(s,a) - V(s) \right\}, \end{aligned} \tag{S11}$$

where

$$\mathcal{T} V(s,a) = \mathbb{E}_{\mathcal{P}(s'|s,a)} V(s').$$

Moreover, the optimal primal solution reads

$$\frac{d_z^\star(s,a)}{d_O(s,a)} = \mathrm{softmax} \left( R_z^\lambda(s,a) + \gamma \mathcal{T} V_z^\star(s,a) - V_z^\star(s) \right). \tag{S12}$$

### C.2  SKILL DISCRIMINATOR TRAINING

With fixed skill-conditioned policy $\pi_z^\star$ and Lagrange multipliers $\lambda > 0$, the Problem S9 becomes

$$\max_{q(z|s)} \sum_z \left\{ \mathbb{E}_{d_z(s,a)} \left[ R_z^\lambda(s,a) \right] - \mathrm{D}_{\mathrm{KL}} \left( d_z(S,A) || d_O(S,A) \right) \right\}$$

and reduces to

$$\max_{q(z|s)} \mathbb{E}_{p(z)} \mathbb{E}_{d_z(s,a)} \log q(z|s).$$

**Lemma C.1.** *Given ratios $\eta_z(s,a)$, using weighted-importance sampling, we can train offline an optimal skill-discriminator $q(z|s)$. In particular, we optimize by gradient descent the following optimization problem*

$$\max_{q(z|s)} \mathbb{E}_{p(z)} \mathbb{E}_{d_O(s,a)} \left[ \eta_z(s,a) \log q(z|s) \right].$$

*Proof.* The statement follows by combining Lem. C.2 and Lem. C.3. $\qquad\square$

**Lemma C.2** (Discriminator Gradient). *It holds that*

$$\nabla_\phi \mathbb{E}_{p(s)} \left[ \mathrm{D}_{\mathrm{KL}} \left( p(Z|s) || q_\phi(Z|s) \right) \right] = -\mathbb{E}_{p(z)} \mathbb{E}_{p(s|z)} \left[ \nabla_\phi \log q_\phi(z|s) \right].$$

*Proof.* Observe that

$$\begin{aligned} \nabla_\phi \mathrm{D}_{\mathrm{KL}} \left( p(Z|s) || q(Z|s) \right) &= \nabla_\phi \mathbb{E}_{p(z|s)} \log \frac{p(z|s)}{q_\phi(z|s)} \\ &= -\mathbb{E}_{p(z|s)} \nabla_\phi \log q_\phi(z|s), \end{aligned}$$

where the second equality follows by

$$\nabla_\phi \log \frac{p(z|s)}{q_\phi(z|s)} = -\frac{q_\phi(z|s)}{p(z|s)} p(z|s) \frac{\nabla_\phi q_\phi(z|s)}{[q_\phi(z|s)]^2} = -\frac{\nabla_\phi q_\phi(z|s)}{q_\phi(z|s)} = -\nabla_\phi \log q_\phi(z|s).$$

$\qquad\square$

**Lemma C.3** (Importance Sampling). *Given ratios $\eta_z(s, a)$, it holds for any function $f(s)$ that*

$$\mathbb{E}_{d_z^\star(s)}\left[f(s)\right] = \mathbb{E}_{d_O(s)}\left[\eta_z(s, a)f(s)\right].$$

*Proof.* Observe that

$$
\begin{aligned}
\mathbb{E}_{d_z^\star(s)}\left[f(s)\right] &= \mathbb{E}_{d_z^\star(s)\pi_z^\star(a|s)}\left[f(s)\right] = \mathbb{E}_{d_z^\star(s,a)}\left[f(s)\right] \\
&= \mathbb{E}_{d_O(s,a)}\left[\eta_z(s, a)f(s)\right].
\end{aligned}
$$

$\square$

## C.3 Estimating State KL Constraint Violation

**Lemma C.4** (State-Action KL Estimator). *Suppose we are given offline datasets $\mathcal{D}_O(S, A) \sim d_O$, $\mathcal{D}_E(S) \sim d_E$ and optimal ratios $\eta_z(s, a) = \frac{d_z(s,a)}{d_O(s,a)}$ and $\eta_{\widetilde{E}}(s, a) = \frac{d_{\widetilde{E}}(s,a)}{d_O(s,a)}$ for all $(s, a) \in \mathcal{D}_O$, where the state-action occupancy $d_{\widetilde{E}}$ is induced by a policy $\pi_{\widetilde{E}}$ agreeing on the state occupancy of an expert $\pi_E$, i.e.*

$$\pi_{\widetilde{E}} \in \arg\min_\pi \mathrm{D}_{\mathrm{KL}}\left(d_\pi(S)||d_E(S)\right).$$

*Then, we can compute* offline *an estimator of $\mathrm{D}_{\mathrm{KL}}\left(d_z(S, A)||d_{\widetilde{E}}(S, A)\right)$ which is given by*

$$\phi_z = \mathbb{E}_{d_O(s,a)}\left[\eta_z(s, a)\log\frac{\eta_z(s, a)}{\eta_{\widetilde{E}}(s, a)}\right].$$

*Proof.* By Lemma C.5 we have

$$\mathrm{D}_{\mathrm{KL}}\left(d_z(S, A)||d_{\widetilde{E}}(S, A)\right) = \mathrm{D}_{\mathrm{KL}}\left(d_z(S, A)||d_O(S, A)\right) - \mathbb{E}_{d_z(s,a)}\left[\log\frac{d_{\widetilde{E}}(s,a)}{d_O(s,a)}\right].$$

For the first term, we have

$$
\begin{aligned}
\mathrm{D}_{\mathrm{KL}}\left(d_z(S, A)||d_O(S, A)\right) &= \mathbb{E}_{d_z(s,a)}\log\frac{d_z(s,a)}{d_O(s,a)} \\
&= \mathbb{E}_{d_O(s,a)}\left[\eta_z(s, a)\log\eta_z(s, a)\right].
\end{aligned}
$$

The second term reduces to

$$\mathbb{E}_{d_z(s,a)}\left[\log\frac{d_{\widetilde{E}}(s,a)}{d_O(s,a)}\right] = \mathbb{E}_{d_O(s,a)}\left[\eta_z(s, a)\log\eta_{\widetilde{E}}(s, a)\right].$$

$\square$

**Lemma C.5** (Structural). *Suppose $0 < \eta_z(s, a), \eta_{\widetilde{E}}(s, a) < \infty$ for all $(s, a) \in \mathcal{D}_O$. Then, we have*

$$\mathrm{D}_{\mathrm{KL}}\left(d_z(S, A)||d_{\widetilde{E}}(S, A)\right) = \mathrm{D}_{\mathrm{KL}}\left(d_z(S, A)||d_O(S, A)\right) - \mathbb{E}_{d_z(s,a)}\left[\log\frac{d_{\widetilde{E}}(s,a)}{d_O(s,a)}\right].$$

*Proof.* By definition of KL-divergence, we have

$$
\begin{aligned}
\mathrm{D}_{\mathrm{KL}}\left(d_z(S, A)||d_{\widetilde{E}}(S, A)\right) &= \mathbb{E}_{d_z(s,a)}\left[\log\left(\frac{d_z(s,a)}{d_O(s,a)} \cdot \frac{d_O(s,a)}{d_{\widetilde{E}}(s,a)}\right)\right] \\
&= \mathrm{D}_{\mathrm{KL}}\left(d_z(S, A)||d_O(S, A)\right) - \mathbb{E}_{d_Z(s,a)}\left[\log\frac{d_{\widetilde{E}}(s,a)}{d_O(s,a)}\right].
\end{aligned}
$$

$\square$

# D  UNCONSTRAINED FORMULATION

SMODICE (Ma et al., 2022) minimizes a KL-divergence between the policy state occupancy and the expert state occupancy, expressed as

$$\min_{d(S)} D_{KL}\left(d(S)||d_E(S)\right). \tag{S13}$$

A naive approach to extend the above problem formulation to the unsupervised skill discovery setting, is to consider an additional diversity term in the objective. In particular, adding a scaled mutual information term $\mathcal{I}(S; Z)$ and maximizing over a set of skill-conditioned state occupancies $\{d_z(S)\}_{z \in Z}$, namely

$$\max_{\{d_z(S)\}_{z \in Z}} \alpha \mathcal{I}(S; Z) - \sum_{z \in Z} D_{KL}\left(d_z(S)||d_E(S)\right). \tag{S14}$$

Here, the level of diversity is controlled by a hyperparameter $\alpha$. However, $\alpha$ is arbitrary, and no constraint on closeness to the expert state occupancy is enforced. We proceed by using the variational lower bound in eq. (3) and assuming a categorical uniform distribution $p(z)$ over the set of latent skills $Z$, which consists of $|Z|$ distinct indicator vectors in $\mathbb{R}^{|Z|}$. This reduce the optimization problem to

$$\max_{d_z(s), q(z|s)} \sum_{z \in Z} \left\{ \alpha \mathbb{E}_{d_z(s)}\left[\frac{\log\left(q(z|s)|Z|\right)}{|Z|}\right] - D_{KL}\left(d_z(S)||d_E(S)\right) \right\}. \tag{S15}$$

**Theorem D.1.** *(Ma et al., 2022) Suppose Assum. 3.1 holds. Then, we have*

$$D_{KL}\left(d_z(S)\|d_E(S)\right) \leq \mathbb{E}_{d_z(s)}\left[\log\frac{d_O(s)}{d_E(s)}\right] + D_{KL}(d_z(S, A)\|d_O(S, A)).$$

By Thm. D.1 and linearity of the objective, Problem (S15) reduces to optimizing separately for each latent skill $z$ the following optimization problem

$$\max_{d_z(s), q(z|s)} \mathbb{E}_{d_z(s)}\left[R_z^\alpha(s, a)\right] - D_{KL}(d_z(S, A)\|d_O(S, A)), \tag{S16}$$

where $R_z^\alpha(s, a)$ is defined as

$$R_z^\alpha(s, a) := \underbrace{\log\frac{d_E(s)}{d_O(s)}}_{\text{Expert Imitation}} + \alpha \underbrace{\frac{\log\left(q(z|s)|Z|\right)}{|Z|}}_{\text{Skill Diversity}}. \tag{S17}$$

The ratios $\frac{d_E(s)}{d_O(s)}$ can be computed by training a discriminator $c(s)$ tasked to distinguish between samples from $d_E(s)$ and $d_O(s)$. More specifically, since the optimal Bayesian discriminator satisfies $c^\star(s) = d_E(s)/(d_E(s) + d_O(s))$, in practice we can use an estimator $c(s)/(1 - c(s)) \approx \frac{d_E(s)}{d_O(s)}$.

Similar to the DOI, we can apply the alternating optimization scheme, here with two phases:(i) fixed skill-discriminator (similarly to Subsec. 4.2.1); and (ii) fixed importance ratios and policy $\pi_z^\star$, where we train the skill-discriminator $q(z|s)$ (see Supp. C.2). For the first phase, we use the importance ratios $\eta_z(s, a)$ computed by optimizing the dual-value problem and then applying softmax to the corresponding TD error terms (see eq. (12) and Nachum and Dai (2020); Ma et al. (2022)).

# E    SOLO-12 DATASET COLLECTION

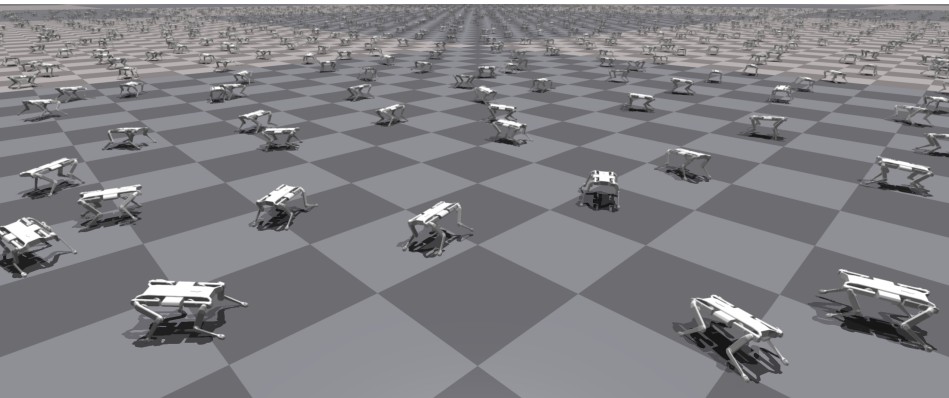

Figure S1: Solo-12 datasets are collected with 4000 environments in parallel using IsaacGym.

As shown in Figure S1, both *expert dataset* and *offline dataset* are collected using locomotion policies trained to track certain velocity in Isaac Gym (Makoviychuk et al., 2021). The policies are trained using an on-policy version of DOMiNO (Zahavy et al., 2022) to exhibit diverse behaviors while maintaining a certain level of tracking. Even trained with randomly sampled velocity, the policies are fed with forward velocity of 1 m/s when collecting both datasets. Both datasets contain 4000 trajectories with an episode length of 250 steps, or 1 million transitions each.

We summarize the main ideas of the training procedure, for details see (Zahavy et al., 2022). Using DOMiNO, we train policies that are conditioned on discrete skill latents and present different behaviors across different skills. Each skill-conditioned policy has a designated skill which is trained with only extrinsic reward and is maintained as the target in the constraint formulation in (Zahavy et al., 2022). We use this target skill from the last training checkpoint (iteration 2000) as the expert of our method. For each skill-conditioned policy, all skills except the target, are trained to balance between extrinsic and intrinsic reward, so as to generate diverse behaviours while being aligned to some degree to the target skill, i.e., maintaining a certain level of tracking velocity. The intrinsic reward is designed to maximize the $\ell_2$ distance of the successor features (Barreto et al., 2016) between different skills, where in our setting the feature space includes: the base height velocity, base roll and pitch velocities, and feet height velocities.

We collected the *offline dataset* using these skill-conditioned policy from different checkpoints during training. The *offline dataset* is composed of 1/2 data from checkpoint 0, 1/4 data from checkpoint 50, 1/8 data from checkpoint 100, 1/16 data from checkpoint 500, 1/32 data from checkpoint 1500 and 1/32 data from checkpoint 2000. For each policy checkpoint, we collect data from the 5 corresponding skills, including the target skill. It is worth noting that more than half of the data from the *offline dataset* comes from the nearly random policies from the start of the training (checkpoint 0 and 50).

Furthermore, in the data collection process, we use a deterministic policy for the *expert dataset*, while for the *offline dataset* we use a stochastic policy. Randomizing the action selection in the latter case, results in more diverse interactions with the environment. In addition, we use domain randomization during training and data collection, in order to tackle the sim-to-real transfer and to simulate more diverse environment interaction. Specifically, we randomize the friction coefficient between $[0.5, 1.5]$ and additional base mass between $[-0.5, 0.5]$ kg, as well as simulate the observation noise and an actuator lag of 15 ms.

# F    SMODICE EXPERT RETURN

In table S1 we show the performance of the evaluated policies trained by SMODICE(Ma et al., 2022) on the WALKER2D and HALFCHEETAH. The results are consistent with the performance that we obtain with DOI in Figure 5. We also note here the importance of having expert state coverage in the offline data that is reflected in the performance of the policies.

| Environment | dataset | $N$ | $r$ |
|---|---|---|---|
| halfcheetah | medium-expert | 25 | 81.25 |
| | | 50 | 80.47 |
| | | 200 | 73.56 |
| | medium-replay | 25 | 29.28 |
| | | 50 | 36.73 |
| | | 200 | 60.67 |
| | random | 25 | 10.89 |
| | | 50 | 27.71 |
| | | 200 | 78.94 |
| walker2d | medium-expert | 25 | 3.98 |
| | | 50 | 19.22 |
| | | 200 | 4.10 |
| | medium-replay | 25 | 15.09 |
| | | 50 | 3.60 |
| | | 200 | 0.95 |
| | random | 25 | 52.62 |
| | | 50 | 103.52 |
| | | 200 | 108.20 |

Table S1: Expected return for SMODICE-learned expert policies in the WALKER2D and ANT environments for $N$ expert trajectories mixed-in.

## G  LAGRANGE MULTIPLIER STABILITY

In Figure S2 we observe the behavior of the Lagrange multipliers for different levels of $\epsilon$ for a specific skill $z$ in the SOLO12 experiment. In case of $\epsilon \in \{1.0, 2.0\}$, the multipliers fluctuate around a specific level that strikes the balance between diversity and expert imitation. This can also be validated when observing the violation level in Figure S2b of the constraint given estimator $\phi_z$, which is for $\epsilon \in \{1.0, 2.0\}$ around 0. On the other hand, if we introduce a strong constraint on the KL-divergence ($\epsilon = 0.0$), which is constantly violated, hence $\sigma(\mu_z) = 1$. Similarly, if the constraint is too weak, only diversity is optimized, in which case there is a significant degradation in performance (see figure Figure 4).

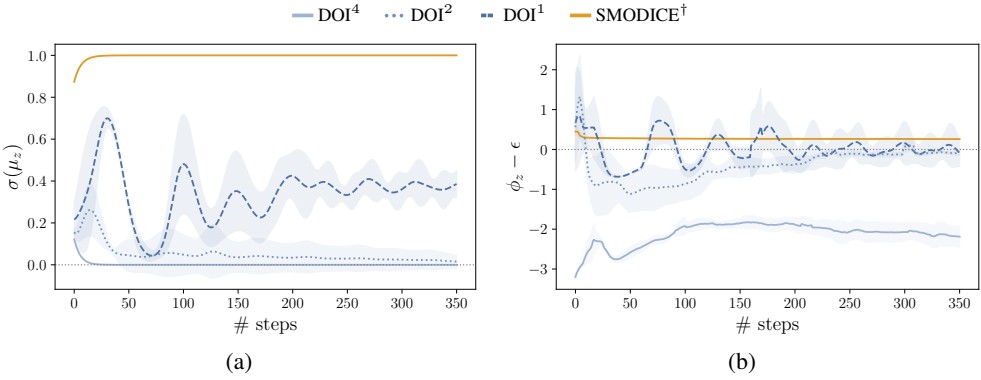

Figure S2: Behavior of Lagrange multipliers. (a) Evolution of $\sigma(\lambda_z)$ for one skill ($z = 1$ chosen arbitrarily), (b) violation of the constraint for different $\epsilon$. Negative $\phi_z - \epsilon$ indicates no violation. Means and standard deviation across restarts.

In Figure S3 we show the bounded lagrange multiplier values for three skills and the resulting violations for different $\epsilon$ levels for the ANT experiment. Again, the multiplier values fluctuate around appropriate levels ensuring the the violation of the constraint remains close to 0.

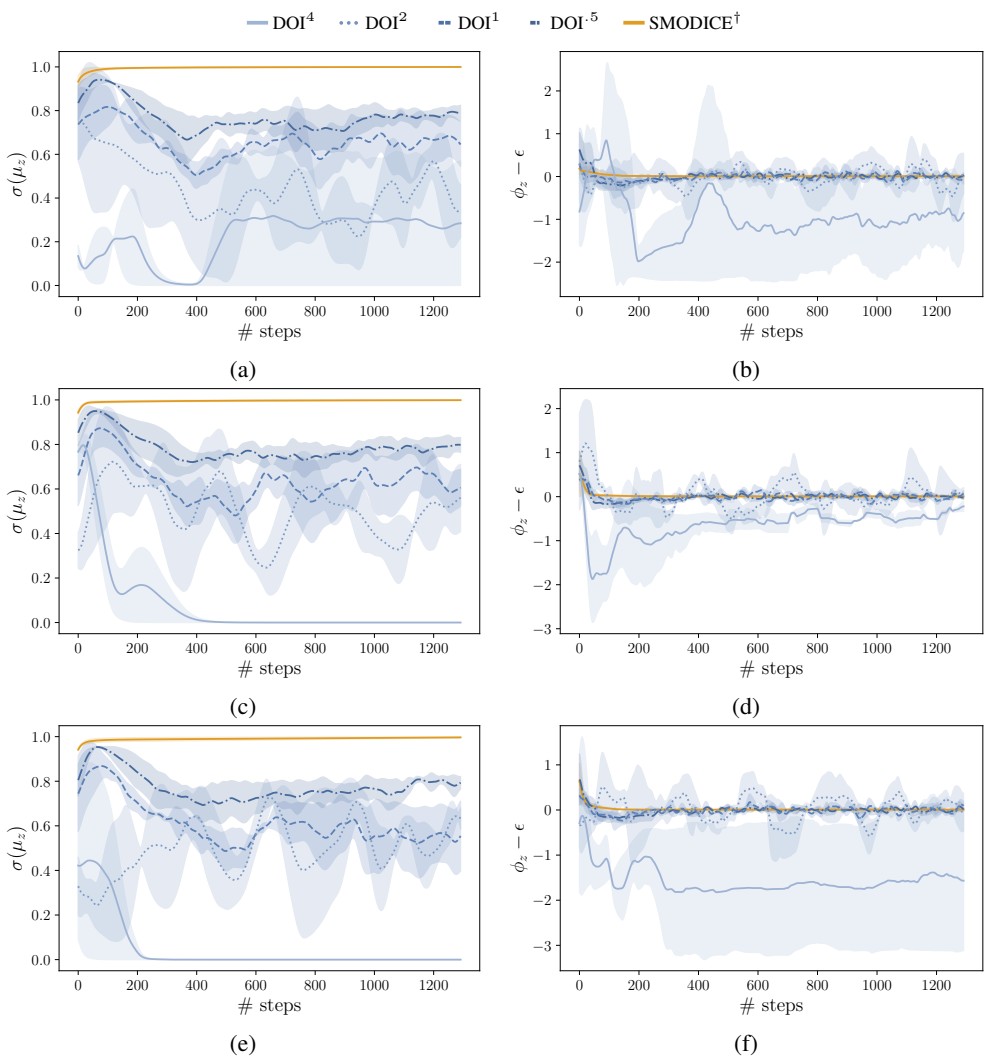

Figure S3: Behavior of Lagrange multipliers. (a) Evolution of $\sigma(\lambda_z)$ for one skill ($z = 1$ chosen arbitrarily), (b) violation of the constraint for different $\epsilon$. Negative $\phi_z - \epsilon$ indicates no violation. Means and standard deviation across restarts.

## H    REAL ROBOT DEPLOYMENT

We successfully deployed policies exhibiting diverse skills extracted from the *offline dataset* while being able to track a certain velocity similar to the expert on real hardware. Our skill-conditioned policy exhibits different walking behaviors with diverse base motions. Snapshots of these diverse behaviors can be seen in Figure S4.

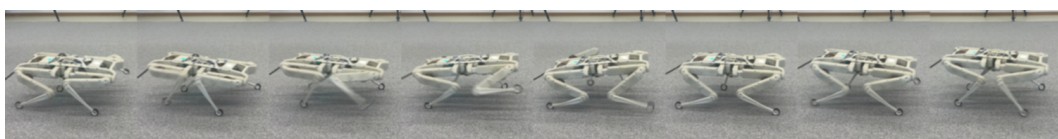

(a) Trot locomotion with wave trunk motion and low base height.

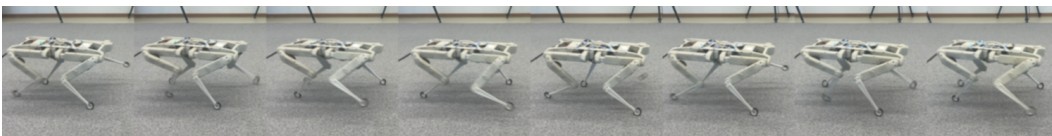

(b) Trot locomotion with middle base height.

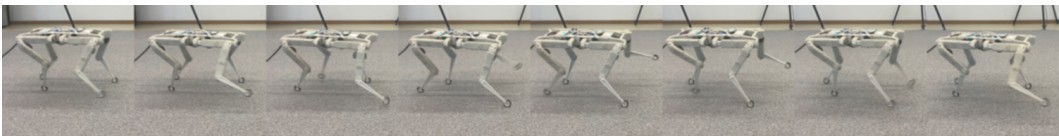

(c) Trot locomotion with high base height.

Figure S4: Snapshots of the trained policy exhibiting different skills on hardware. From above to bottom, the policy has low, middle and high base positions while moving forward.

## I    OBSERVATION PROJECTION

Imitation learning is of particular interest when the agent's and the target expert policy's state spaces do not necessarily match, but overlap in certain parts, as is often the case when learning from demonstrations. Our framework naturally accounts for this. If we consider $\mathcal{S}'$ to be the state space of the expert and $\mathcal{S}$ the state space of the agent, we assume that there exists a simple projection mapping $\Pi : \mathcal{S}' \mapsto \mathcal{O}$, where $\mathcal{O} := \{o : o \subset s, s \in \mathcal{S}\}$ is the power set of observations, allowing us to potentially imitate beyond expert policies with the same state space as the agent. Note that the agent still observes its full state $s$, however the projected state $\Pi(s)$ is observed by the expert classifier and skill discriminator. The projection $\Pi$ can be selected to specify which parts of the state we want to diversify and constrain in terms of occupancy, depending on the task at hand.

## J    LIMITATIONS

The DOI method also comes with certain caveats. Maximizing the mutual information, as a diversity objective, poses a hard optimization problem due to its convexity. Thus, designing alternative diversity objectives can be beneficial. Furthermore, closeness in state-action occupancy can be quite restrictive in terms of availability of diverse behaviors that satisfy the constraint. Replacing this with constraints on the return of the policy would allow more freedom to optimize diversity in cases where the optimal policy may be multimodal. The above challenges are promising directions for future work.

## K ADDITIONAL EXPERIMENTS

Instead of learning the Lagrange multipliers $\lambda_z$ via KL estimators $\phi_z$, we can also fix $\lambda_z$ at a certain level, making it a hyperparameter. In our setting, this also works well, and we demonstrate a tradeoff between diversity and task reward optimization, see Figures S5 and S6. However, in this case we lose the possiblity to enforce a certaint constraint on the KL-divergence between the skill state-action occupancy and expert state-action occupancy.

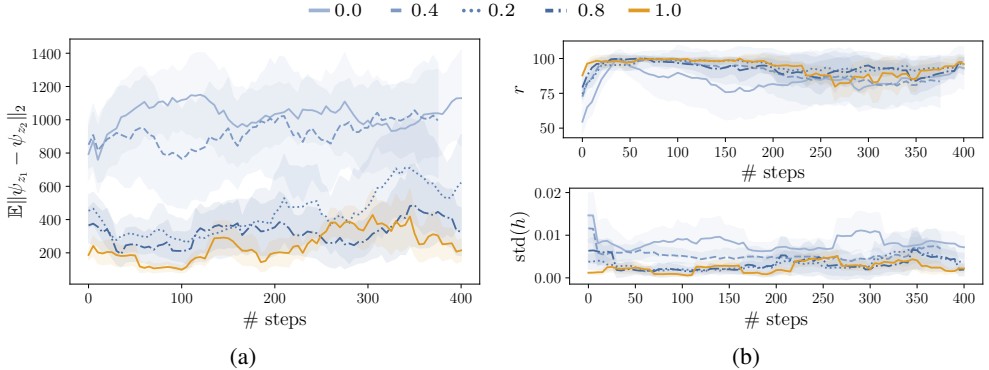

(a)                                         (b)

Figure S5: (a) Average $\ell_2$ distance between Monte Carlo estimated successor representations $\psi_z$ of different skills, (b) return $r$ as % of expert return and standard deviation of base height $\mathrm{std}_z(h)$, depending on a fixed $\sigma(\lambda_z)$ (see legend).

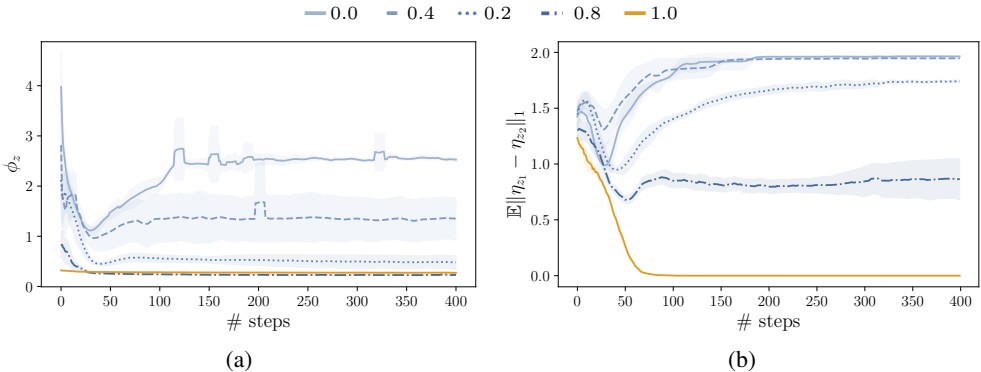

(a)                                         (b)

Figure S6: Divergence estimate and $\eta_z$ distance for the case of fixed $\sigma(\lambda_z)$. (a) Value of divergence estimator $\phi_z$ for a specific skill over the course of training ($z = 1$ chosen arbitrarily), (b) average $\ell_1$ distance of $\eta_z$'s of skills. Means and standard deviation across restarts.

We further provide results of applying DOI to different levels of expert trajectory mix-in to the *medium-replay* and *random* datasets of WALKER2D and HALFCHEETAH in tables S2 and S3.

| dataset | # expert mixin | $\epsilon$ | $\mathbb{E}\|\eta_{z_1} - \eta_{z_2}\|$ | $r$ | $\mathbb{E}\|\psi_{z_1} - \psi_{z_2}\|$ |
|---|---|---|---|---|---|
| medium-replay | 25 | 0.0 | $0.00 \pm 0.00$ | $46.00 \pm 1.46$ | $6.16 \pm 0.30$ |
| | | 0.5 | $0.21 \pm 0.08$ | $0.33 \pm 0.48$ | $3.54 \pm 2.14$ |
| | | 1.0 | $1.40 \pm 0.05$ | $2.33 \pm 0.51$ | $6.09 \pm 2.40$ |
| | | 2.0 | $1.30 \pm 0.03$ | $0.64 \pm 0.11$ | $7.67 \pm 4.27$ |
| | | 4.0 | $1.54 \pm 0.08$ | $2.30 \pm 1.64$ | $19.26 \pm 2.29$ |
| | 50 | 0.0 | $0.00 \pm 0.00$ | $54.29 \pm 2.13$ | $5.53 \pm 0.14$ |
| | | 0.5 | $0.82 \pm 0.28$ | $31.31 \pm 7.03$ | $14.13 \pm 1.86$ |
| | | 1.0 | $1.21 \pm 0.15$ | $4.33 \pm 0.75$ | $0.42 \pm 0.05$ |
| | | 2.0 | $1.37 \pm 0.03$ | $1.61 \pm 0.41$ | $13.85 \pm 2.50$ |
| | | 4.0 | $1.48 \pm 0.12$ | $1.11 \pm 0.36$ | $22.02 \pm 1.33$ |
| | 200 | 0.0 | $0.00 \pm 0.00$ | $98.33 \pm 0.44$ | $2.67 \pm 0.26$ |
| | | 0.5 | $0.45 \pm 0.11$ | $74.59 \pm 8.96$ | $6.22 \pm 1.17$ |
| | | 1.0 | $1.20 \pm 0.09$ | $2.52 \pm 1.50$ | $12.97 \pm 4.33$ |
| | | 2.0 | $1.30 \pm 0.03$ | $2.07 \pm 0.65$ | $3.23 \pm 2.02$ |
| | | 4.0 | $1.59 \pm 0.06$ | $1.43 \pm 0.64$ | $19.48 \pm 1.43$ |
| random | 25 | 0.0 | $0.00 \pm 0.00$ | $36.49 \pm 11.54$ | $15.70 \pm 0.48$ |
| | | 0.5 | $0.93 \pm 0.02$ | $20.48 \pm 7.90$ | $16.81 \pm 3.14$ |
| | | 1.0 | $1.30 \pm 0.12$ | $3.72 \pm 1.38$ | $8.16 \pm 5.43$ |
| | | 2.0 | $1.45 \pm 0.09$ | $1.22 \pm 0.32$ | $20.47 \pm 3.08$ |
| | | 4.0 | $1.27 \pm 0.05$ | $0.60 \pm 0.26$ | $20.60 \pm 4.17$ |
| | 50 | 0.0 | $0.00 \pm 0.00$ | $103.16 \pm 0.69$ | $3.32 \pm 0.07$ |
| | | 0.5 | $1.03 \pm 0.13$ | $33.60 \pm 6.64$ | $18.27 \pm 2.50$ |
| | | 1.0 | $1.37 \pm 0.09$ | $5.05 \pm 2.66$ | $20.16 \pm 3.05$ |
| | | 2.0 | $1.46 \pm 0.06$ | $0.77 \pm 0.29$ | $10.46 \pm 3.77$ |
| | | 4.0 | $1.23 \pm 0.09$ | $0.26 \pm 0.11$ | $14.33 \pm 1.97$ |
| | 200 | 0.0 | $0.00 \pm 0.00$ | $107.43 \pm 0.26$ | $1.84 \pm 0.08$ |
| | | 0.5 | $1.29 \pm 0.07$ | $103.29 \pm 1.38$ | $6.75 \pm 0.77$ |
| | | 1.0 | $1.26 \pm 0.22$ | $2.43 \pm 0.30$ | $7.30 \pm 4.86$ |
| | | 2.0 | $1.46 \pm 0.10$ | $0.47 \pm 0.15$ | $15.39 \pm 1.56$ |
| | | 4.0 | $1.29 \pm 0.01$ | $1.91 \pm 0.57$ | $19.66 \pm 3.36$ |

Table S2: WALKER2D metrics across random and medium-replay variants with varying number of mixed-in trajectories of the expert to satisfy the coverage assumption.

| dataset | # expert mixin | $\epsilon$ | $\mathbb{E}\|\eta_{z_1} - \eta_{z_2}\|$ | $r$ | $\mathbb{E}\|\psi_{z_1} - \psi_{z_2}\|$ |
|---|---|---|---|---|---|
| medium-replay | 25 | 0.0 | $0.00 \pm 0.00$ | $37.64 \pm 0.30$ | $3.22 \pm 0.06$ |
| | | 0.5 | $0.83 \pm 0.12$ | $36.95 \pm 0.63$ | $3.02 \pm 0.10$ |
| | | 1.0 | $1.36 \pm 0.09$ | $24.30 \pm 6.28$ | $13.34 \pm 4.84$ |
| | | 2.0 | $1.44 \pm 0.06$ | $6.73 \pm 3.65$ | $22.09 \pm 8.15$ |
| | | 4.0 | $1.27 \pm 0.09$ | $2.68 \pm 0.72$ | $21.68 \pm 1.87$ |
| | 50 | 0.0 | $0.01 \pm 0.01$ | $45.40 \pm 0.22$ | $3.26 \pm 0.27$ |
| | | 0.5 | $1.14 \pm 0.02$ | $42.89 \pm 0.19$ | $2.94 \pm 0.12$ |
| | | 1.0 | $1.41 \pm 0.12$ | $37.28 \pm 2.41$ | $6.18 \pm 1.21$ |
| | | 2.0 | $1.32 \pm 0.11$ | $8.60 \pm 4.66$ | $13.66 \pm 1.97$ |
| | | 4.0 | $1.24 \pm 0.16$ | $1.72 \pm 0.18$ | $28.74 \pm 7.84$ |
| | 200 | 0.0 | $0.00 \pm 0.00$ | $73.60 \pm 0.39$ | $3.65 \pm 0.09$ |
| | | 0.5 | $1.16 \pm 0.08$ | $69.91 \pm 1.14$ | $3.67 \pm 0.10$ |
| | | 1.0 | $1.28 \pm 0.13$ | $23.74 \pm 12.94$ | $13.47 \pm 1.73$ |
| | | 2.0 | $1.49 \pm 0.10$ | $15.52 \pm 4.29$ | $32.03 \pm 0.56$ |
| | | 4.0 | $1.42 \pm 0.07$ | $2.16 \pm 0.04$ | $11.92 \pm 2.28$ |
| random | 25 | 0.0 | $0.00 \pm 0.00$ | $2.80 \pm 0.36$ | $5.55 \pm 1.18$ |
| | | 0.5 | $1.12 \pm 0.04$ | $3.03 \pm 0.28$ | $4.30 \pm 0.85$ |
| | | 1.0 | $1.14 \pm 0.12$ | $2.24 \pm 0.09$ | $10.45 \pm 3.30$ |
| | | 2.0 | $1.24 \pm 0.08$ | $1.73 \pm 0.33$ | $25.01 \pm 8.78$ |
| | | 4.0 | $1.44 \pm 0.03$ | $1.60 \pm 0.30$ | $35.08 \pm 8.27$ |
| | 50 | 0.0 | $0.00 \pm 0.00$ | $31.89 \pm 1.14$ | $9.97 \pm 0.58$ |
| | | 0.5 | $1.14 \pm 0.11$ | $10.29 \pm 3.13$ | $17.90 \pm 6.01$ |
| | | 1.0 | $1.42 \pm 0.15$ | $6.45 \pm 2.95$ | $23.30 \pm 0.96$ |
| | | 2.0 | $1.41 \pm 0.08$ | $2.73 \pm 0.43$ | $23.91 \pm 6.98$ |
| | | 4.0 | $1.68 \pm 0.06$ | $1.44 \pm 0.27$ | $35.07 \pm 8.08$ |
| | 200 | 0.0 | $0.00 \pm 0.00$ | $68.35 \pm 1.25$ | $5.20 \pm 0.31$ |
| | | 0.5 | $1.30 \pm 0.08$ | $50.85 \pm 17.30$ | $9.80 \pm 3.68$ |
| | | 1.0 | $1.21 \pm 0.12$ | $15.06 \pm 5.58$ | $29.57 \pm 4.26$ |
| | | 2.0 | $1.03 \pm 0.10$ | $2.10 \pm 1.99$ | $10.84 \pm 7.57$ |
| | | 4.0 | $1.20 \pm 0.20$ | $2.16 \pm 0.05$ | $16.90 \pm 5.95$ |

Table S3: HALFCHEETAH metrics across random and medium-replay variants with varying number of mixed-in trajectories of the expert to satisfy the coverage assumption.

