# OpenReview forum: "Diverse Offline Imitation Learning"
_ICLR.cc/2024/Conference — Submitted to ICLR 2024_

### Official Review · Reviewer_7Hkf · 2023-10-29

**Soundness:** 3 good
**Presentation:** 2 fair
**Contribution:** 2 fair
**Rating:** 3
**Confidence:** 4

**Summary:**

This paper introduces DOI (Diverse Offline Imitation) for improving diversity of learned skill-conditioned policies. These skills are trained to have maximum diversity (via maximizing mutual information between states and skills) and to be close to expert state distributions. Prior work, SMODICE, is an instance of DOI where the KL-divergence between each skill state-visitation distribution and the expert state-visitation distribution must be 0, but DOI relaxes this constraint to be up to $\varepsilon$.

DOI is evaluated on a 12-DOF quadruped both in sim and real as well as in D4RL and shown to promote greater diversity of skills. However, there is a tradeoff between skill diversity and performance.

**Strengths:**

- Method was deployed on a real robot, and website has videos showing the learning of presumably 3 different skills involving locomotion at 3 different base heights.
- Figure 1 was helpful.
- Writing in the main text sections of the paper was mostly clear, and the math was pretty understandable step-by-step.

**Weaknesses:**

## Method, Experiments, and Argument

(A1) Novelty and algorithmic contribution seems quite limited compared to SMODICE. The main differences are (1) Learning a discrete number of skill-based policies instead of a single policy, and (2) allowing skill-based policies and the expert to have up-to-$\varepsilon$ KL divergence. Limited algorithmic novelty is fine if robotic performance on at least a few downstream tasks is a lot better than prior work, but experiments show ultimate performance is not improved with DOI.

(A2) Experimental metrics are not very illustrative of the performance of DOI. Most of the plots (Figure 3, 4a) show proxy measures of how diverse the skills are. Only Figure 4b (and Tables S2, S3) compares DOI performance with SMODICE. Figure 4b shows that performance of DOI at smaller epsilon values is comparable to SMODICE, and Tables S2 and S3 show that SMODICE ($\varepsilon = 0$) does better than any $\varepsilon > 0$. Perhaps this is expected, as the authors argued there is a diversity-performance tradeoff in Section 6. But if so, there should be experiments, such as those testing generalization, where higher skill diversity leads to better robustness than narrowly-learned policies. However, the authors did not have experimental results of this kind, which brings into question when DOI should be used over SMODICE.

(A3) Abstract and conclusion states that this paper proposes “a principled offline algorithm for unsupervised skill discovery.” However, the paper assumes that the set of skills $Z$ is finite, which suggests that skill candidates are predefined and presumably not continuous-spaced. If predefined, where is the “skill discovery” coming from? This is different from how CIC [1] does skill discovery, where skill vectors are continuous and sampled from a prior distribution.


## Presentation/Coherence

Overall, results graphs and algorithm boxes need to be clearer.

(B1) Paper did not precisely define what a skill is, what the set of finite skills $Z$ is initialized to, or where $Z$ is from. Is $z \in Z$ a continuous vector or a one-hot?

(B2) Since method section relies so much on SMODICE, and Figure 2 refers to DICE (which was not introduced until page 7), I would recommend putting the related work section before the preliminaries section.

(B3) Nitpick: Section 3.2.1, first line, should refer to Problem (eqn 7) instead of Problem (eqn 6), I believe.

(B4) Algorithm 1 talks about a discriminator $c^{*}$ mapping state to predicted probability that the state is from $d_E$ vs $d_O$. However, this discriminator is not mentioned anywhere in the main text and can be confused with the skill discriminator $q(z|s)$.

(B5) Phase 2 in Algorithm 1 mentions training $\pi_z^{*}$ when Section 3.2.1 seems to say the policy is trained in Phase 1 instead.

(B6) Section 5, Related Work, reads more like a laundry list of prior papers. There is no comparison at all to this paper’s proposed method, DOI.

(B7) Real robot results are not discussed in Section 6.

(B8) Tables S2 (Walker2D) and S3 (HalfCheetah) contain the exact same entries (at least for the ~20 cells I looked at). This seems like an unfortunate mistake.

## Reference
[1] CIC: Contrastive Intrinsic Control for Unsupervised Skill Discovery. Laskin et al.

**Questions:**

1. How would DOI be extendable to a continuous skill space?

2. How did authors define the skill space $Z$ for each of the different environments? What was $Z$ and $|Z|$ for the D4RL envs? Presumably it was $|Z| = 3$ and $Z = [\text{low base}, \text{mid base}, \text{high base}]$ for locomotion?

3. Given Assumption 2.1, I wonder what the state representation is for the locomotion task. Can DOI be adapted to work on image observations, or a learned image feature space?

4. What is $N$ in Tables S2 and S3?

5. What does the color-coding in Figure 3a mean? Is it the same color meaning as Figure 3b?

6. In Algorithm 1, the reward is defined as function of the output of $c^{*}$, which could easily be confused with the reward terms mentioned in Equations 8 and 14. How is this reward term, which doesn’t depend on actions, used to compute importance sampling ratios $\eta_{\tilde{E}}(s,a)$, which does?

7. Figure 3a: x-axis is confusingly labeled “data.” Section 6 describes this as “across the dataset assignment.” What do these things mean?

8. I suggest using a slightly more descriptive title for the paper.

---

> ### Author Response · Authors · 2023-11-22
>
> > (A1) Novelty and algorithmic contribution seems quite limited compared to SMODICE. The main differences are (1) Learning a discrete number of skill-based policies instead of a single policy, and (2) allowing skill-based policies and the expert to have up-to-$\epsilon$ KL divergence. Limited algorithmic novelty is fine if robotic performance on at least a few downstream tasks is a lot better than prior work, but experiments show ultimate performance is not improved with DOI.
>
> We respectfully disagree with the reviewer's comment.
> It is crucial to emphasize that the novelty of our work is two-fold:
> i) we propose a novel and practically relevant problem formulation; and
> ii) we show that combining existing algorithmic techniques yields a principled offline algorithm.
> We want to stress the importance of understanding that previous work on unsupervised skill discovery reduces maximizing diversity to online training of a policy and a skill discriminator.
> Although there are efficient offline RL algorithms for learning a policy that maximizes a fixed reward, they cannot be applied here because our constraint optimization setting involves a non-stationary reward that depends on: i) the log-likelihood of the skill discriminator; and ii) the Lagrange multipliers.
> In addition, training the skill discriminator and estimating the KL divergence both offline remains a challenge for non-DICE-based algorithms.
> In summary, we extend DICE-based offline imitation to diversity maximization with imitation constraints, by utilizing the importance-weighted occupancy ratios to train a skill discriminator and to estimate KL divergence offline.
> We have added this explanation more clearly in the paper.
>
> > (A2) Experimental metrics are not very illustrative of the performance of DOI. Most of the plots (Figure 3, 4a) show proxy measures of how diverse the skills are. Only Figure 4b (and Tables S2, S3) compares DOI performance with SMODICE. Figure 4b shows that performance of DOI at smaller epsilon values is comparable to SMODICE, and Tables S2 and S3 show that SMODICE ($\epsilon=0$) does better than any $\epsilon>0$. Perhaps this is expected, as the authors argued there is a diversity-performance tradeoff in Section 6. But if so, there should be experiments, such as those testing generalization, where higher skill diversity leads to better robustness than narrowly-learned policies. However, the authors did not have experimental results of this kind, which brings into question when DOI should be used over SMODICE.
>
>
> It is indeed valuable to consider an offline data set consisting of various trajectories starting at $s$ and ending at $t$.
> Let the state-only expert samples be from the shortest path (straight line) between $s$ and $t$.
> SMODICE learns policies that closely follow the straight line. Blocking the straight line with an obstacle causes SMODICE to fail, while DOI learns diverse skills to get around the obstacle. We can provide this for the camera-ready.
>
> Another potential application is to enhance the realism of computer games by creating an immersive experience of interacting with non-player characters, each behaving in a slightly different style, while all partially imitating the behavior of a human expert.
>
> > (A3) Abstract and conclusion states that this paper proposes “a principled offline algorithm for unsupervised skill discovery.” However, the paper assumes that the set of skills is finite, which suggests that skill candidates are predefined and presumably not continuous-spaced. If predefined, where is the “skill discovery” coming from? This is different from how CIC [1] does skill discovery, where skill vectors are continuous and sampled from a prior distribution.
>
> Thank you for raising this important question.
> However, there is a misunderstanding about what the word ``skill'' denotes in the context of unsupervised skill discovery.
> It is standard in the literature to denote by $z$ a vector in $\mathbb{R}^d$ and refer to it as a latent skill.
> We need to emphasize that the latent skills are not learnable, they are fixed.
> In the discrete setting, they are usually chosen to be indicator vectors (there are $d$ distinct latent skills in $\mathbb{R}^d$).
> In contrast, it is the skill-conditioned policy $\pi(a|s,z)$ that is learnable, while the latent skill $z$ serves as an index of a learned temporally extended behavior.
> Moreover, a skill-conditioned policy $\pi\_z$ induces a state occupancy $d\_z(S)$ which belongs to the state simplex.
> Then, the geometric interpretation of maximizing mutual information $\mathcal{I}(S;Z)=\mathbb{E}\_{z}\mathrm{D}\_{\mathrm{KL}}(d\_{z}||\mathbb{E}\_{z^{\prime}}d\_{z^{\prime}})$ is to learn $d$ many points $d_{z}(S)$ such that the distance between each point and the mean point is maximized.
> In practice, each skill-conditional policy $\pi\_z$ maximizes a reward given by the log-likelihood of a learnable skill-discriminator $q(z|s)$.
> We have clarified and updated the paper accordingly.

---

> ### Author Response · Authors · 2023-11-22
>
> > Overall, results graphs and algorithm boxes need to be clearer.
>
> We have clarified and updated the paper accordingly.
>
> > (B1) Paper did not precisely define what a skill is, what the set of finite skills $Z$ is initialized to, or where $Z$ is from. Is $z\in Z$ a continuous vector or a one-hot?
>
> Thank you for pointing that out.
> However, in the Preliminaries section we state that we treat $Z$ as a finite set (paragraph 3), and before Eq. (3) we specify $p(z) = \frac{1}{|Z|}$ as the uniform distribution over $Z$.
> Furthermore, it is standard in the literature, when $p(z)$ is a categorical distribution to choose latent skills $z$ as indicator vectors ($|Z|$ distinct skills in $\mathbb{R}^{|Z|}$).
> Nonetheless, following the reviewer's comment, we have added this explanation more clearly in the paper.
>
> > (B2) Since method section relies so much on SMODICE, and Figure 2 refers to DICE (which was not introduced until page 7), I would recommend putting the related work section before the preliminaries section.
>
> Following the reviewer's comment, we have updated the paper accordingly.
>
> > (B3) Nitpick: Section 3.2.1, first line, should refer to Problem (eqn 7) instead of Problem (eqn 6), I believe.
>
> Thank you for pointing that out.
> We have updated the paper accordingly.
>
> > (B4) Algorithm 1 talks about a discriminator $c^\star$ mapping state to predicted probability that the state is from $d_E$ vs $d_O$. However, this discriminator is not mentioned anywhere in the main text and can be confused with the skill discriminator.
>
> Thank you for pointing that out.
> We have updated the paper with a clearer separation between state-discriminator $c^{\star}(s)$ and skill-discriminator $q(z|s)$.
>
> > (B5) Phase 2 in Algorithm 1 mentions training
>  $\pi_z^*$ when Section 3.2.1 seems to say the policy is trained in Phase 1 instead.
>
> Thank you for pointing that out.
> We have updated the paper accordingly.
>
> > (B6) Section 5, Related Work, reads more like a laundry list of prior papers. There is no comparison at all to this paper’s proposed method, DOI.
>
> Thank you for your valuable suggestion.
> We have refined the "Related Work" section to emphasize the differences between our work and the existing literature and to improve overall clarity.
>
> > (B7) Real robot results are not discussed in Section 6.
>
> It is important to note that due to space limitations, we have deferred further discussion of the real robot results to the Supplementary Material (F, G, H).
> However, the data collection for training the policy was discussed in Section 6.
>
> > (B8) Tables S2 (Walker2D) and S3 (HalfCheetah) contain the exact same entries (at least for the ~20 cells I looked at). This seems like an unfortunate mistake.
>
> We thank the reviewer for noticing this, this was indeed a mistake and we corrected it in the revision.

---

> ### Author Response · Authors · 2023-11-22
>
> > Q1. How would DOI be extendable to a continuous skill space?
>
> Thank you for raising this important question.
> The short answer is: by replacing the uniform categorical distribution $p(z)$ over the set $Z$ of indicator vectors in $\mathbb{R}^{|Z|}$, with a continuous uniform distribution over the surface of a unit ball $\mathcal{S}^{d-1}$ in $\mathbb{R}^d$.
> However, it is important to realize that the skill-discriminator $q:\mathcal{S}\rightarrow\triangle_{\mathcal{S}^{d-1}}$ now maps states to continuous probability distribution over the latent skill space $\mathcal{S}^{d-1}$, and computing this probability distribution is prohibitively expensive in practice.
> While overcoming this challenge in our setting is an interesting open question, it is important to emphasize that addressing it is beyond the scope of this paper.
>
> > Q2. How did authors define the skill space  for each of the different environments? ...
>
> $Z$ is a discrete set of $|Z|$ many distinct indicator vectors in $\mathbb{R}^{|Z|}$.
> More specifically, $|Z|=5$ for all environments.
> For Solo12, this is shown in Figure 3, each color corresponds to a different skill.
>
> > Q3. What does $N$ mean in Tables S2 and S3?
>
> $N$ is the number of expert trajectories that are mixed-in into the offline dataset (but not labeled), as done in SMODICE.
>
> > Q4. What does the color-coding in Figure 3a mean? Is it the same color meaning as Figure 3b?
>
> Each skill is a different color.
> Figures 3a and 3b show that the importance ratios of different skills are evaluated on one million data points (which are sub-sampled).
>
> > Q5. In Algorithm 1, the reward is defined as function of the output of $c^{*}$, which could easily be confused with the reward terms mentioned in Equations 8 and 14. How is this reward term, which doesn’t depend on actions, used to compute importance sampling ratios $\eta\_{\tilde{E}}(s,a)$, which does?
>
> Thank you for raising this important question.
> The short answer is: applying Phase 1 with reward $R(s,a)$ is in fact the call to the SMODICE algorithm, cast into our more general framework.
> Intuitively, the reward $R(s,a)=\log\frac{c^{\star}(s)}{1-c^{\star}(s)}$ does not depend on the action, since the objective in SMODICE is to compute a state occupancy $d\_z(S)$ that minimizes the KL-divergence to an expert state occupancy $d\_E(S)$.
> For a detailed discussion, see Section Unconstrained Formulation in Appendix D.
> Nonetheless, following the reviewer's comment, we updated Algorithm 1 to make it clearer that in Phase 1, the Value function $V\_{z}^{\star}$ is optimized with respect to the reward $R\_{z}^{\mu}(s,a)$ which is defined in Eq. (15).
>
> > Q6. Figure 3a: x-axis is confusingly labeled “data.” Section 6 describes this as “across the dataset assignment.” What do these things mean?
>
> Thank you for pointing that out.
> Along the x-axis, we plot the state-action $(s,a)$ pairs from the offline dataset.
> The figures illustrate a clustering effect on the data induced by the discovered skills.
>
> > Q7. I suggest using a slightly more descriptive title for the paper.
>
> Following the reviewer's suggestion, we updated the title to: ``DOI: Offline Diversity Maximization under Imitation Constraints''.

---

### Official Review · Reviewer_iADi · 2023-10-31

**Soundness:** 3 good
**Presentation:** 2 fair
**Contribution:** 2 fair
**Rating:** 3
**Confidence:** 3

**Summary:**

The paper addresses the question of learning diverse skills from offline dataset while being close to expert. The objective is formulated and existing ideas are used to solve the combined objective effectively. The resulting method is tested on offline datasets from D4RL and a real robot to demonstrate the diversity induced by their method.

**Strengths:**

1. The paper presents a new objective for unsupervised skill learning from offline datasets - maximize mutual information combined with staying close to expert. To facilitate this, the method leverages algorithms from previous works that are off-policy in nature. To solve the mutual information objective they use the variational lower bound previously seen in DIAYN, to optimize the KL-constraint they use DICE method and for optimizing objective and constraint jointly the lagrangian method is used, commonly seen in safe RL literature. The combination results in a new off-policy skill discovery method.
2.  The paper empirically demonstrates effectiveness of their method on simulated tasks on D4RL by comparing metrics for diversity using the offline dataset. They show the tradeoff of contraint vs diversity and the expected difference of importance ratio.
3. The paper also test the algorithm on a real quaduped which learns gaits with various heights.

**Weaknesses:**

1. The paper misses to explain the motivation behind proposing the objective:

Why stay close to expert? If the objective is to generate diverse skill, what is the objective of incoporating the constraint of staying close to expert. An explanation through examples might help to motivate the paper better.
KL divergence to expert: Any objective that uses KL divergence is extremely sensitive if the learned skill goes out of support of the expert dataset. It is motivated in the paper the skill should imitate some part of expert but this is not what is enforced by the KL divergence.

2. Theoretical contributions: I believe the lemma’s are minor variations over previous works in DICE space [1,2,3,4] which might be discussed and compared to more thoroughly.
3. Evaluation:
    1. Online evaluation: The paper currently only plots metrics on offline dataset. I believe this is not the correct metric. A learned visitation distribution might not be practically feasible although theoretically it should be. One way to test it in simulated domains, is to roll out pi_z for different skills and compare the resulting visitation distribution.
    2. Qualitative Diversity of skill: An important part of the evaluation process should be a qualitative comparison of the skills the algorithm learns. If the algorithm learn meaningless skills it would be clear from the deployed policy.
    3. Baselines and Quantitative diversity of skill: No prior methods for skill discovery are compared against. A standard comparison might be the resulting estimated mutual information of skills between prior methods and DOI. Although prior methods are not developed for offline setting, a simple extension would be to pair them with offline RL. Example: DIAYN could be combined with IQL instead of SAC.

**Questions:**

1. In the paragraph after Figure 3, How does skill conditioned variant of SMODICE does not have a discriminator?  SMODICE itself learns a discriminator in the original method which estimates ratio of expert to offline data. Do you mean the skill discriminator?
2. I am not sure how to compute the expectation of importance ratio for different skills. Is the expectation over the offline dataset?
3. For SOLO12, the data is already generated by a skill-based algorithm which seems their is a strong prior to recovering the same skills as the original algorithm? Can you ablate how the learned skills are different from skills found by Domino?

---

> ### Author Response · Authors · 2023-11-22
>
> > 1. The paper misses to explain the motivation behind proposing the objective: Why stay close to expert? If the objective is to generate diverse skill, what is the objective of incorporating the constraint of staying close to expert. An explanation through examples might help to motivate the paper better.
>
> Thank you for raising this important question.
> A central topic in computer game development is how to create an authentic real-world experience using a rich set of non-player characters.
> Our work makes a step towards achieving this goal, by proposing a principled framework that captures the core challenge.
> Given inexpensive and easily collectable expert state demonstrations, our algorithm generates a diverse set of skills that imitate to some degree the expert.
> In this way, each skill behaves slightly differently while visiting a similar state trajectory as the expert, meaning that it behaves similarly to the expert or, more broadly, mimics the expert's style.
> Crucially, our algorithm learns the sequence of actions that lead to these behaviors offline!
> We have updated the paper accordingly.
>
> > 1.1 KL divergence to expert: Any objective that uses KL divergence is extremely sensitive if the learned skill goes out of support of the expert dataset. It is motivated in the paper the skill should imitate some part of expert but this is not what is enforced by the KL divergence.
>
> Thank you for the insightful comment.
> In fact, our offline estimator of the KL divergence, uses ratios $\eta_{z}(s,a)$ and $\eta_{\widetilde{E}}(s,a)$ that are computed only on state-action pairs within the offline dataset $\mathcal{D}_O$.
> Furthermore, in practice, we ensure that these ratios are strictly positive, so that the KL estimator $\phi_z$ is well defined and bounded.
> Although for simplicity of presentation we choose to state our algorithm in terms of KL divergence, as done in previous work, all of our results easily generalize to f-divergence.
> In particular, for the camera-ready version, we will present an additional set of experiments with $\chi^2$ divergence and report them in the appendix.
>
> > 2. Theoretical contributions: I believe the lemma’s are minor variations over previous works in DICE space [1,2,3,4] which might be discussed and compared to more thoroughly.
>
> Thank you for your valuable suggestion.
> We have updated section "4.2 Approximation Phases" accordingly.
>
> > 3.1 Online evaluation: The paper currently only plots metrics on offline dataset. I believe this is not the correct metric. A learned visitation distribution might not be practically feasible although theoretically it should be. One way to test it in simulated domains, is to roll out $\pi_z$ for different skills and compare the resulting visitation distribution.
>
> Thank you for the insightful suggestion.
> In fact, we consider the expected successor feature distance in $\ell_2$ norm over an initial state distribution, which is an online Monte Carlo estimate (policy rollout) that we specify on page 8 and include as an evaluation metric for both Solo12 and D4RL environments.
>
> > 3.2 Qualitative Diversity of skill: An important part of the evaluation process should be a qualitative comparison of the skills the algorithm learns. If the algorithm learn meaningless skills it would be clear from the deployed policy.
>
> Thank you for your valuable suggestion.
> However, the skills were qualitatively evaluated on the Solo12, see Appendix G, and videos were provided on the website.

---

> ### Author Response · Authors · 2023-11-22
>
> > 3.3 Baselines and Quantitative diversity of skill: No prior methods for skill discovery are compared against. A standard comparison might be the resulting estimated mutual information of skills between prior methods and DOI. Although prior methods are not developed for offline setting, a simple extension would be to pair them with offline RL. Example: DIAYN could be combined with IQL instead of SAC.
>
> Thank you for raising this important question.
> The short answer is: there is no baseline in the offline setting.
> It is crucial to emphasize the unique characteristics of our problem formulation.
> More specifically, we consider offline RL imitation given state-only expert demonstrations, while seeking to maximize the diversity of imitating skills.
> First, an extensive body of work on DICE-based algorithms has shown that offline RL imitation based on Fenchel duality achieves state-of-the-art results on the task of imitating an expert from state-only demonstrations.
> In our setting, however, in addition to the constraint that each skill imitates an expert (in the above sense), we seek to maximize the skill diversity.
> Second, previous work on unsupervised skill discovery reduces maximizing diversity to online training of a policy and a skill discriminator.
> It is worth noting that while there are efficient offline RL algorithms for learning a policy that maximizes a fixed reward, they cannot be applied here because our constraint optimization setting involves a non-stationary reward that depends on: i) the log-likelihood of the skill discriminator; and ii) the Lagrange multipliers.
> In addition, training the skill discriminator and estimating the KL divergence both offline remains a challenge for non-DICE-based algorithms.
> In summary, we extend DICE-based offline imitation to diversity
> maximization with imitation constraints, by utilizing the importance-weighted
> occupancy ratios to train a skill discriminator and to estimate KL divergence
> offline.
> We have added this explanation more clearly in the paper.
>
> > Q1. In the paragraph after Figure 3, How does skill conditioned variant of SMODICE does not have a discriminator? SMODICE itself learns a discriminator in the original method which estimates ratio of expert to offline data. Do you mean the skill discriminator?
>
> Thank you for pointing that out.
> The short answer is yes, we mean the skill discriminator.
> In more details: although SMODICE trains a state discriminator $c(s)$, it is important to realize that in our context, the state discriminator $c(s)$ is encapsulated in the preprocessing call to SMODICE.
> That is, our algorithm has access only to the returned importance ratios $\eta_{\widetilde{E}}(s,a)$.
> Each skill of SMODICE$^{\dagger}$ maximizes the reward function $\log\eta_{\widetilde{E}}(s,a)$ without using the skill discriminator $q(z|s)$, as it only seeks to imitate without diversifying, $\sigma(\mu_z)=1$.
> We have updated the paper accordingly.
>
> > Q2. I am not sure how to compute the expectation of importance ratio for different skills. Is the expectation over the offline dataset?
>
> Thank you for raising this important question.
> It is indeed valuable to understand how the DICE-based framework is used to design an offline expectation estimation scheme.
> Our goal is to compute offline primal optimal ratios $\eta\_z(s,a)=d\_z^{\star}(s,a)/d\_O(s,a)$ of state-action occupancies with respect to a fixed reward for each state-action pair $(s,a)$ in the offline dataset $\mathcal{D}\_O$, see Problem (10).
> To achieve this, we solve offline the dual problem that yields an optimal Value function $V^{\star}$, see Eq. (11).
> A fundamental theorem in Fenchel duality states that given the optimal Value function $V^{\star}$, the primal optimal ratios $\eta_z(s,a)$ admit a close-form solution, i.e., softmax of the TD error w.r.t. $V^{\star}$, see Eq. (12).
> These ratios $\eta_z(s,a)$ are then used to design an offline importance-weighted sampling procedure that, for an arbitrary function $f$, satisfies $\mathbb{E}\_{d\_{z}^{\star}(s,a)} [f(s,a,z)] = \mathbb{E}\_{d_{O}(s,a)}[\eta\_{z}(s,a) f(s,a,z)]$.
> Then, the optimal skill-conditioned policy $\pi\_z^{\star}$ is trained offline using a weighted behavioral cloning, which is obtained by setting $f(s,a,z)=\log(\pi\_z(a|s))$.
> Furthermore, we can train offline the skill discriminator by setting $f(s,a,z)=\log(q(z|s))$ (see Lemma 4.2), and we can estimate the KL divergence by setting $f(s,a,z)=\log( \eta\_z(s,a)/\eta_{\widetilde{E}}(s,a) )$ (see Lemma 4.3).

---

### Official Review · Reviewer_gvyu · 2023-11-01

**Soundness:** 3 good
**Presentation:** 3 good
**Contribution:** 3 good
**Rating:** 5
**Confidence:** 3

**Summary:**

This paper proposes a novel algorithm to connect Fenchel duality, reinforcement learning, and unsupervised skill discovery to maximize a mutual information objective subject to KL-divergence state occupancy constraints. This approach is used to diversify offline policies for a 12-DoF quadruped robot and several environments from the standard D4RL benchmark in terms of both ℓ2 distance of expected successor features and ℓ1 distance of importance ratios.

**Strengths:**

The strength of this paper is that it proposes a principled offline algorithm for unsupervised skill discovery that maximizes diversity while ensuring each learned skill imitates state-only expert demonstrations to a certain degree. In order to compute the optimal solution to the problem formulation, the authors propose to use an approximation algorithm "alternative optomization". The authors demonstrate the effectiveness of the method on standard offline benchmarks and a custom offline dataset collected from a quadruped robot. The resulting skill diversity naturally entails a trade-off in task performance, which can be controlled via a KL constraint level ϵ.

**Weaknesses:**

1. In the experiment section, it will be good to see some comparison between the proposed method with other state-of-the-art methods for unsupervised skill discovery.
2. Computational complexity of the proposed algorithm is not mentioned in the paper.
3. The paper does not provide a comprehensive evaluation of the proposed method on a wide range of tasks and environments.

**Questions:**

Is there a convergence analysis of the algorithm? What if the algorithm does not converge?

---

> ### Author Response · Authors · 2023-11-22
>
> > 1. In the experiment section, it will be good to see some comparison between the proposed method with other state-of-the-art methods for unsupervised skill discovery.
>
> Thank you for raising this important question.
> The short answer is: there is no baseline in the offline setting.
> It is crucial to emphasize the unique characteristics of our problem formulation.
> More specifically, we consider offline RL imitation given state-only expert demonstrations, while seeking to maximize the diversity of imitating skills.
> First, an extensive body of work on DICE-based algorithms has shown that offline RL imitation based on Fenchel duality achieves state-of-the-art results on the task of imitating an expert from state-only demonstrations.
> In our setting, however, in addition to the constraint that each skill imitates an expert (in the above sense), we seek to maximize the skill diversity.
> Second, previous work on unsupervised skill discovery reduces maximizing diversity to online training of a policy and a skill discriminator.
> It is worth noting that while there are efficient offline RL algorithms for learning a policy that maximizes a fixed reward, they cannot be applied here because our constraint optimization setting involves a non-stationary reward that depends on: i) the log-likelihood of the skill discriminator; and ii) the Lagrange multipliers.
> In addition, training the skill discriminator and estimating the KL divergence both offline remains a challenge for non-DICE-based algorithms.
> In summary, we extend DICE-based offline imitation to diversity maximization with imitation constraints, by utilizing the importance-weighted
> occupancy ratios to train a skill discriminator and to estimate KL divergence offline.
> We have added this explanation more clearly in the paper.
>
> > 2. Computational complexity of the proposed algorithm is not mentioned in the paper.
>
> Thank you for raising this important remark.
> Maximizing mutual information (a convex function) is generally intractable, and as a trackable surrogate we consider instead a variational lower bound.
> It is standard in previous work to use an alternating scheme heuristic that involves two phases: a policy and a skill discriminator optimization.
> This heuristic does not provide computational complexity or convergence guarantees.
> In addition, the non-stationarity of the reward signal in our setting also depends on Lagrange multipliers, which makes the resulting three-phase optimization problem more challenging.
>
> > 3. The paper does not provide a comprehensive evaluation of the proposed method on a wide range of tasks and environments.
>
> Our experiments on Solo12 and D4RL environments strongly support the effectiveness of our novel problem formulation.
> By simultaneously addressing the challenges of state-only expert imitation and skill diversity maximization, our proposed algorithm is the first principled offline approach for this novel setting.
>
> > 4. Is there a convergence analysis of the algorithm? What if the algorithm does not converge?
>
> To our understanding reviewer's comment 2) and this one are closely related.
> It is crucial to emphasize that the commonly used alternating optimization scheme is a heuristic and does not provide computational complexity or convergence guarantee.
> In practice, the algorithm converges to some local optimum.

---

### Author Response · Authors · 2023-11-23

We would like to thank the reviewers for the time they spent reviewing our paper and for their valuable comments.

Several reviewers recognized the novelty of the setting studied, as well as the applicability of our algorithm to practical problems.
More specifically:
- (gvyu) "proposes a principled offline algorithm'', "demonstrate the effectiveness of the method'';
- (iADi) "presents a new objective", "new off-policy skill discovery method", "empirically demonstrates effectiveness", "test the algorithm on a real quaduped";
- (7Hkf) "method was deployed on a real robot", "clear writing", "math was pretty understandable".

Thank you for the positive feedback.

The main concerns raised relate to i) lack of comparison with other state-of-the-art methods (gvyu, iADi); and ii) the motivation behind the proposed problem formulation and method (iADi, 7Hkf).

- i) It is crucial to emphasize that, to the best of our knowledge, there are no offline baselines for unsupervised skill discovery; and
- ii) Previous work has focused on the online setting and only recently considered a constraint formulation that quantitatively measures the utility of each skill by maximizing a reward function.
While mathematically convenient, this is a strong assumption, as reward engineering is a challenging task.
In this work, we require access to state-only samples of expert demonstrations, without knowledge of the expert actions.
This setting is particularly valuable in robotics scenarios where expert state-action demonstrations are limited and the domain of the expert may be different from that of the agent, such as in human demonstrations.
Furthermore, building upon DICE-based off-policy techniques, we propose the first principled offline algorithm for maximizing skill diversity under imitation constraints.

Our work makes a step towards learning offline a set of robust skills for reachability tasks.
Another potential application is to enhance the realism of computer games by creating an immersive experience of interacting with non-player characters, each behaving in a slightly different style, while all partially imitating the behavior of a human expert.

We believe our answers and changes detailed in the individual responses clarify the questions and address the raised concerns about our work. Changes to the paper are highlighted in violet in the revision. We hope that the changes that we have made and the responses will encourage the reviewers to update their scores.

---

### Meta-Review · Area_Chair_FjTv · 2023-12-06

**Metareview:**

(a) Summary
The paper presents a novel algorithm for unsupervised skill discovery in an offline setting. The authors propose a method that integrates Fenchel duality, reinforcement learning, and unsupervised skill discovery to maximize a mutual information objective while adhering to KL-divergence state occupancy constraints. Their approach aims to diversify offline policies by learning skills that imitate state-only expert demonstrations to some degree. Key findings include the effectiveness of the method in diversifying skills for a 12-DoF quadruped robot and various environments from the D4RL benchmark, as evidenced by ℓ2 and ℓ1 distance metrics.

(b) Strengths:
(+) Novel Algorithmic Approach (Reviewer gvyu, iADi, 7Hkf): The paper introduces a principled offline algorithm for unsupervised skill discovery. It is commended for creatively combining existing techniques in a novel way, addressing a unique problem formulation in the offline setting.
(+) Practical Relevance and Applicability (Reviewer iADi, 7Hkf): The method’s applicability to real-world scenarios, particularly in robotics, is a significant strength. Its deployment on a real quadruped robot and demonstration on D4RL benchmarks highlight its practical relevance.
(+) Clear Presentation and Understandable Mathematics (Reviewer 7Hkf): The paper is clearly written. The mathematical aspects of the approach are laid out in an understandable manner.

(c) Weaknesses
(-) Limited Comparison with State-of-the-Art Methods (Reviewer gvyu, iADi): A major weakness is the lack of comparison with other state-of-the-art methods in unsupervised skill discovery, particularly in the offline setting. This makes it challenging to gauge the proposed method's relative performance and innovation.
(-) Concerns Over Online Evaluation (Reviewer iADi): The paper’s online evaluation metrics and methods are questioned for their adequacy in capturing the performance of the learned skills. There's a need for more comprehensive and illustrative metrics to evaluate the skills practically.
(-) Algorithmic Novelty and Contribution (Reviewer 7Hkf): There are concerns regarding the extent of the algorithmic novelty compared to existing methods like SMODICE. The distinction and advancements over these methods are not clear.
(-) Theoretical and Practical Contributions (Reviewer iADi, 7Hkf): The paper’s theoretical contributions are seen as minor variations over previous works in the DICE space, and the practical implications of the learned skills require further elucidation.
(-) Lack of Detailed Implementation and Convergence Analysis (Reviewer gvyu, iADi): The paper does not thoroughly discuss the computational complexity and convergence analysis of the proposed algorithm, which are crucial for understanding its feasibility and reliability.

Other missing components:
* More detailed discussion on the scalability and computational demands of the proposed method
* More diversified test environments or scenarios to demonstrate the robustness and versatility of the proposed method.

**Justification For Why Not Higher Score:**

The paper fails to clear the bar on sufficient novelty over prior work like SMODICE. This, in addition to a lack of metrics for evaluating the learnt skills, pushes it below the bar for accept.

**Justification For Why Not Lower Score:**

N/A

---

### Decision · Program_Chairs · 2024-01-16

Reject